# A Unified Framework for Reinforcement Learning under Policy and Dynamics Shifts

## Abstract

Training reinforcement learning policies using environment interaction data collected from varying policies or dynamics presents a fundamental challenge. Existing works often overlook the distribution discrepancies induced by policy or dynamics shifts, or rely on specialized algorithms with task priors, thus often resulting in suboptimal policy performances and high learning variances. In this paper, we identify a unified strategy for online RL policy learning under diverse settings of policy and dynamics shifts: transition occupancy matching. In light of this, we introduce a surrogate policy learning objective by considering the transition occupancy discrepancies and then cast it into a tractable *min-max* optimization problem through dual reformulation. Our method, dubbed Occupancy-Matching Policy Optimization (OMPO), features a specialized actor-critic structure and a distribution discriminator. We conduct extensive experiments based on the OpenAI Gym, Meta-World, and Panda Robots environments, encompassing policy shifts under stationary and non-stationary dynamics, as well as domain adaption. The results demonstrate that OMPO outperforms the specialized baselines from different categories in all settings. We also find that OMPO exhibits particularly strong performance when combined with domain randomization, highlighting its potential in RL-based robotics applications.

## 1 Introduction

Online Reinforcement Learning (RL) aims to learn policies to maximize long-term returns through interactions with the environments, which has achieved significant advances in recent years (Gu et al., 2017; Bing et al., 2022b; Mnih et al., 2013; Perolat et al., 2022; Cao et al., 2023). Many of these advances rely on on-policy data collection, wherein agents gather fresh experiences in stationary environments to update their policies (Schulman et al., 2015; 2017; Zhang & Ross, 2021). However, this on-policy approach can be expensive or even impractical in some real-world scenarios, limiting its practical applications. To overcome this limitation, a natural cure is to enable policy learning with data collected under varying policies or dynamics (Haarnoja et al., 2018; Rakelly et al., 2019; Duan et al., 2021; Zanette, 2023; Xue et al., 2023).

Challenges arise when dealing with such collected data with policy or dynamics shifts, which often diverge from the distribution induced by the current policy under the desired target dynamics. Naïvely incorporating such shifted data during training without careful identification and treatment, could lead to erroneous policy evaluation (Thomas & Brunskill, 2016; Irpan et al., 2019), eventually resulting in biased policy optimization (Imani et al., 2018; Nota & Thomas, 2020; Chan et al., 2022). Previous methods often only focus on specific types of policy or dynamics shifts, lacking a unified understanding and solution to address the underlying problem. For example, in stationary environments, off-policy methods such as those employing importance weights or off-policy evaluation have been used to address policy shifts (Jiang & Li, 2016; Fujimoto et al., 2018; Zanette & Wainwright, 2022). Beyond policy shifts, dynamics shifts can occur in settings involving environment or task variations, which are common in task settings such as domain randomization (Tobin et al., 2017; Chen et al., 2021; Kadokawa et al., 2023), domain adaptation (Eysenbach et al., 2021; Liu et al., 2021), and policy learning under non-stationary environments (Rakelly et al., 2019; Lee et al., 2020; Wei & Luo, 2021; Bing et al., 2022a). According to different combinations of dynamics and policy shifts, we categorize these different scenarios into three types: *1) policy shifts with*

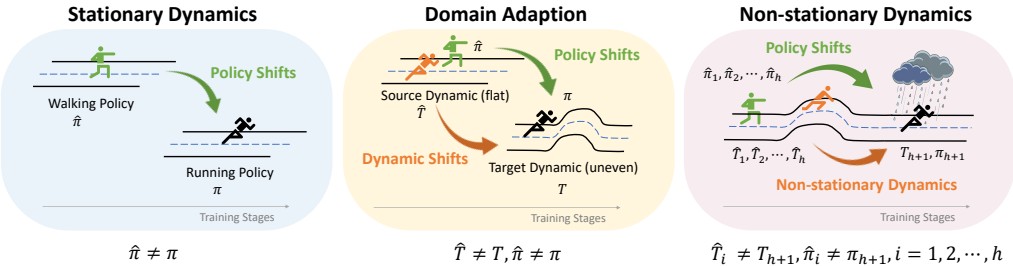

Figure 1: Diverse settings of online reinforcement learning under policy or dynamics shifts.

*stationary dynamics*, *2) policy shifts with domain adaption*, and *3) policy shifts with non-stationary dynamics* (see Figure 1 for an intuitive illustration[1]).

Our work stems from a realization that, from a distribution perspective, under the same state $s$, policy shifts lead to different choices of actions $a$, while dynamics shifts primarily introduce discrepancies over the next states $s'$ given state-action pair $(s, a)$. Regardless of the combination of policy and dynamics shifts, the discrepancies inherently boil down to the transition occupancy distribution involving $(s, a, s')$. This means that if we can correct the transition occupancy discrepancies among data from various sources during the RL training process, *i.e.*, implementing transition occupancy matching, we can elegantly model all policy & dynamics shift scenarios within a unified framework.

Inspired by this insight, we introduce a novel and unified framework, Occupancy-Matching Policy Optimization (OMPO), designed to facilitate policy learning with data affected by policy and dynamic shifts. We start by presenting a surrogate policy objective capable of capturing the impacts of transition occupancy discrepancies. We then show that this policy objective can be transformed into a tractable *min-max* optimization problem through dual reformulation (Nachum et al., 2019b), which naturally leads to an instantiation with a special actor-critic structure and a distribution discriminator.

We conduct extensive experiments on diverse benchmark environments to demonstrate the superiority of OMPO, including locomotion tasks from OpenAI Gym (Brockman et al., 2016) and manipulation tasks in Meta-World (Yu et al., 2019) and Panda Robots (Gallouédec et al., 2021) environments. Our results show that OMPO can achieve consistently superior performance using a single framework as compared to prior specialized baselines from diverse settings. Notably, when combined with domain randomization, OMPO exhibits remarkable performance gains and sample efficiency improvement, which makes it an ideal choice for many RL applications facing sim-to-real adaptation challenges, *e.g.*, robotics tasks.

## 2 RELATED WORKS

We first briefly summarize relevant methods that handle diverse types of policy and dynamics shifts.

**Policy learning under policy shifts with stationary dynamics.** In scenarios involving policy shifts, several off-policy RL methods have emerged that leverage off-policy experiences stored in the replay buffer for policy evaluation and improvement (Jiang & Li, 2016; Fujimoto et al., 2018; Zanette & Wainwright, 2022; Ji et al., 2023). However, these approaches either ignore the impact of policy shifts or attempt to reconcile policy gradients through importance sampling (Precup, 2000; Munos et al., 2016). Unfortunately, due to off-policy distribution mismatch and function approximation error, these methods often suffer from high learning variance and training instability, potentially hindering policy optimization and convergence (Nachum et al., 2019b).

**Policy learning under policy shifts with domain adaption.** Domain adaptation scenarios involve multiple time-invariant dynamics, where policy training is varied in the source domain to ensure the resulting policy is adaptable to the target domain. Methods in this category typically involve modifying the reward function to incorporate target domain knowledge (Arndt et al., 2020; Eysenbach et al., 2021; Liu et al., 2021). However, these methods often require the source domain dynamics can cover the target domain dynamics, and potentially involve extensive human design to achieve

---

[1]The walking and running pictograms are from `https://olympics.com/en/sports/`

optimal performance. Another setting is domain randomization, where the source domains are randomized to match the target domain (Tobin et al., 2017; Chen et al., 2021; Kadokawa et al., 2023). Nonetheless, these techniques heavily rely on model expressiveness and generalization, which do not directly address shifts in policy and time-invariant dynamics.

**Policy learning under policy shifts with non-stationary dynamics.** Non-stationary dynamics encompass a broader class of environments where dynamics can change at any timestep, such as encountering unknown disturbances or structural changes. Previous works have often adopted additional latent variables to infer possible successor dynamics (Lee et al., 2020; Wei & Luo, 2021; Bing et al., 2022a), learned the stationary state distribution from data with dynamics shift (Xue et al., 2023), or improved dynamics model for domain generalization (Cang et al., 2021). However, these methods either rely on assumptions about the nature of dynamics shifts, such as hidden Markov models (Bouguila et al., 2022) and Lipschitz continuity (Domingues et al., 2021), or neglect the potential policy shifts issues, limiting their flexibility across non-stationary dynamics with policy shifts settings.

## 3 PRELIMINARIES

We consider the typical Markov Decision Process (MDP) setting (Sutton & Barto, 2018), which is denoted by a tuple $\mathcal{M} = \langle \mathcal{S}, \mathcal{A}, r, T, \mu_0, \gamma \rangle$. Here, $\mathcal{S}$ and $\mathcal{A}$ represent the state and action spaces, while $r : \mathcal{S} \times \mathcal{A} \to (0, r_{\max}]$ is the reward function. The transition dynamics $T : \mathcal{S} \times \mathcal{A} \to \Delta(\mathcal{S})$ captures the probability of transitioning from state $s_t$ and action $a_t$ to state $s_{t+1}$ at timestep $t$. The initial state distribution is represented by $\mu_0$, and $\gamma \in (0, 1]$ is the discount factor. Given an MDP, the objective of RL is to find a policy $\pi : \mathcal{S} \to \Delta(\mathcal{A})$ that maximizes the cumulative reward obtained from the environment, which can be formally expressed as $\pi^* = \arg\max_\pi \mathbb{E}_{s_0 \sim \mu_0, a \sim \pi(\cdot|s), s' \sim T(\cdot|s,a)} \left[ \sum_{t=0}^{\infty} \gamma^t r(s_t, a_t) \right]$.

In this paper, we employ the dual form of the RL objective (Puterman, 2014; Wang et al., 2007; Nachum et al., 2019b), which can be represented as follows:

$$\pi^* = \arg\max_\pi \mathcal{J}(\pi) = \arg\max_\pi \mathbb{E}_{(s,a)\sim\rho^\pi} \left[ r(s, a) \right]. \tag{1}$$

where $\rho^\pi(s, a)$ represents a normalized discounted state-action occupancy distribution (henceforth, we omit "normalized discounted" for brevity), characterizing the distribution of state-action pairs $(s, a)$ induced by policy $\pi$ under the dynamics $T$. It can be defined as:

$$\rho^\pi(s, a) = (1 - \gamma) \sum_{t=0}^{\infty} \gamma^t \Pr\left[ s_t = s, a_t = a | s_0 \sim \mu_0, a_t \sim \pi(\cdot|s_t), s_{t+1} \sim T(\cdot|s_t, a_t) \right].$$

To tackle this optimization problem, a class of methods known as the DIstribution Correction Estimation (DICE) has been developed (Nachum et al., 2019b; Lee et al., 2021; Kim et al., 2021; Ma et al., 2022; 2023; Li et al., 2022). These methods leverage offline data or off-policy experience to estimate the on-policy distribution $\rho^\pi(s, a)$, and subsequently learn the policy. In this paper, we extend DICE-family methods to the transition occupancy matching context (see more discussion in Appendix D), addressing the challenges posed by policy and dynamics shifts in a unified framework.

## 4 POLICY OPTIMIZATION UNDER POLICY AND DYNAMICS SHIFTS

We consider the online off-policy RL setting, where the agent interacts with environments, collects new experiences $(s, a, s', r)$ and stores them in a replay buffer $\mathcal{D}$. At each training step, the agent samples a random batch from $\mathcal{D}$ to update the policy. We use $\widehat{\pi}$ and $\widehat{T}$ to denote the historical/source policies and dynamics specified by the replay buffer (Hazan et al., 2019; Zhang et al., 2021), while $\pi$ and $T$ to denote the current policy and target/desired dynamics. For the aforementioned three policy & dynamics shifted types, we have

- **Policy shifts with stationary dynamics**: Only policy shifts occur ($\widehat{\pi} \neq \pi$), while the dynamics remain stationary[2] ($\widehat{T} \simeq T$).

---

[2]We use "$\simeq$" to represent that the empirical dynamics derived from sampling data can be approximately equal to the true dynamics.

- ***Policy shifts with domain adaption***: Both policy shifts ($\widehat{\pi} \neq \pi$) and gaps between the source and target dynamics ($\widehat{T} \neq T$) can be observed.
- ***Policy shifts with non-stationary dynamics***: Policy shifts ($\widehat{\pi} \neq \pi$) occur alongside dynamics variation ($\widehat{T}_1, \widehat{T}_2, \cdots, \widehat{T}_h \neq T_{h+1}$). For simplicity, we consider a mixture of historical dynamics in the replay buffer, representing this as ($\widehat{T} \neq T$).

Through estimating the discrepancies between different state-action distributions, the mismatch between $\rho_\pi(s,a)$ and $\rho_{\widehat{\pi}}(s,a)$ can be effectively corrected for policy shifts. However, when both policy and dynamics shifts occur, using state-action occupancy alone, without capturing the next state $s'$ for dynamics shifts, is insufficient. To address this, we introduce the concept of transition occupancy distribution (Viano et al., 2021; Ma et al., 2023). This distribution considers the normalized discounted marginal distributions of state-actions pair $(s,a)$ as well as the next states $s'$:

$$\rho_T^\pi(s,a,s') = (1-\gamma) \sum_{t=0}^{\infty} \gamma^t \Pr\left[s_t = s, a_t = a, s_{t+1} = s' | s_0 \sim \mu_0, a_t \sim \pi(\cdot|s_t), s_{t+1} \sim T(\cdot|s_t, a_t)\right].$$

Hence, the policy & dynamics shifts in the previous three types can be generalized as transition occupancy discrepancies, *i.e.*, $\rho_T^\pi(s,a,s') \neq \rho_{\widehat{T}}^{\widehat{\pi}}(s,a,s')$. This offers a new opportunity for developing a unified modeling framework to handle diverse policy and dynamics shifts. In this section, we propose a surrogate policy learning objective that captures the transition occupancy discrepancies, which can be further cast into a tractable *min-max* optimization problem through dual reformulation.

## 4.1 A Surrogate Policy Learning Objective

With the transition occupancy distribution $\rho_T^\pi(s,a,s')$ in hand, we redefine the policy learning objective as $\mathcal{J}(\pi) = \mathbb{E}_{(s,a,s') \sim \rho_T^\pi}[r(s,a)]$. Employing the fact $x > \log(x)$ for $x > 0$ and Jensen's inequality, we provide the following policy learning objective:

$$\begin{aligned}
\mathcal{J}(\pi) > \log \mathcal{J}(\pi) &= \log \mathbb{E}_{(s,a,s') \sim \rho_T^\pi}[r(s,a)] = \log \mathbb{E}_{(s,a,s') \sim \rho_{\widehat{T}}^\pi}\left[\left(\rho_T^\pi / \rho_{\widehat{T}}^\pi\right) \cdot r(s,a)\right] \\
&\geq \mathbb{E}_{(s,a,s') \sim \rho_{\widehat{T}}^\pi}\left[\log\left(\rho_T^\pi / \rho_{\widehat{T}}^\pi\right) + \log r(s,a)\right] \\
&= \mathbb{E}_{(s,a,s') \sim \rho_{\widehat{T}}^\pi}\left[\log r(s,a)\right] - D_{\mathrm{KL}}\left(\rho_{\widehat{T}}^\pi(s,a,s') \| \rho_T^\pi(s,a,s')\right).
\end{aligned} \quad (2)$$

Here, $D_{\mathrm{KL}}(\cdot)$ represents the KL-divergence that measures the distribution discrepancy introduced by the dynamics $\widehat{T}$. In cases encountering substantial dynamics shifts, the term $D_{\mathrm{KL}}(\cdot)$ can be large, subordinating the reward and causing training instability. Drawing inspiration from prior methods (Haarnoja et al., 2018; Nachum et al., 2019b; Xu et al., 2023), we introduce a weighted factor $\alpha$ to balance the scale and focus on the following more practical objective:

$$\bar{\mathcal{J}}(\pi) = \mathbb{E}_{(s,a,s') \sim \rho_{\widehat{T}}^\pi}\left[\log r(s,a)\right] - \alpha D_{\mathrm{KL}}\left(\rho_{\widehat{T}}^\pi(s,a,s') \| \rho_T^\pi(s,a,s')\right). \quad (3)$$

We further incorporate the policy $\widehat{\pi}$ into this objective to account for policy shifts. The following proposition provides an upper bound for the KL-divergence discrepancy:

**Proposition 4.1.** *Let $\rho_{\widehat{T}}^{\widehat{\pi}}(s,a,s')$ denote the transition occupancy distribution specified by the replay buffer. The following inequality holds for any $f$-divergence that upper bounds the KL divergence:*

$$D_{\mathrm{KL}}\left(\rho_{\widehat{T}}^\pi \| \rho_T^\pi\right) \leq \mathbb{E}_{(s,a,s') \sim \rho_{\widehat{T}}^\pi}\left[\log\left(\rho_T^\pi / \rho_{\widehat{T}}^{\widehat{\pi}}\right)\right] + D_f\left(\rho_{\widehat{T}}^\pi \| \rho_{\widehat{T}}^{\widehat{\pi}}\right). \quad (4)$$

The proof is provided in Appendix A.1. By substituting the bound (4) into objective (3), we can establish a surrogate policy learning objective under policy and dynamics shifts:

$$\widehat{\mathcal{J}}(\pi) = \mathbb{E}_{(s,a,s') \sim \rho_{\widehat{T}}^\pi}\left[\log r(s,a) - \alpha \log \frac{\rho_T^\pi(s,a,s')}{\rho_{\widehat{T}}^{\widehat{\pi}}(s,a,s')}\right] - \alpha D_f\left(\rho_{\widehat{T}}^\pi(s,a,s') \| \rho_{\widehat{T}}^{\widehat{\pi}}(s,a,s')\right). \quad (5)$$

The final surrogate objective involves $\rho_{\widehat{T}}^{\widehat{\pi}}(s,a,s')$, making it theoretically possible to utilize data from the replay buffer for policy learning, and also allowing us to explicitly investigate the impacts of policy and dynamics shifts.

## 4.2 DUAL REFORMULATION OF THE SURROGATE OBJECTIVE

Directly solving the surrogate objective has some difficulties, primarily due to the presence of the unknown distribution $\rho_{\widehat{T}}^\pi$. Estimating this distribution necessitates sampling from the current policy $\pi$ samples in historical dynamics $\widehat{T}$. While some model-based RL methods (Janner et al., 2019; Ji et al., 2022) in principle can approximate such samples through model learning and policy rollout, these approaches can be costly and lack feasibility in scenarios with rapidly changing dynamics. Instead of dynamics approximation, we can rewrite the definition of transition occupancy distribution as the following *Bellman flow* constraint (Puterman, 2014) in our optimization problem,

$$\rho_{\widehat{T}}^\pi(s,a,s') = (1-\gamma)\mu_0(s)\widehat{T}(s'|s,a)\pi(a|s) + \gamma\widehat{T}(s'|s,a)\pi(a|s)\sum_{\hat{s},\hat{a}}\rho_{\widehat{T}}^\pi(\hat{s},\hat{a},s).$$

Let $\mathcal{T}_\star^\pi\rho^\pi(s,a) = \pi(a|s)\sum_{\hat{s},\hat{a}}\rho_{\widehat{T}}^\pi(\hat{s},\hat{a},s)$ denote the transpose (or adjoint) transition operator. Note that $\sum_{s'}\rho_T^\pi(s,a,s') = \rho^\pi(s,a)$ and $\sum_{s'}T(s'|s,a) = 1$, we can integrate over $s'$ to remove $\widehat{T}$:

$$\rho^\pi(s,a) = (1-\gamma)\mu_0(s)\pi(a|s) + \gamma\mathcal{T}_\star^\pi\rho^\pi(s,a), \quad \forall(s,a)\in\mathcal{S}\times\mathcal{A}. \tag{6}$$

Thus, to enable policy learning with the surrogate objective, we seek to solve the following equivalent constrained optimization problem:

$$\pi^* = \arg\max_\pi \widehat{\mathcal{J}}(\pi) = \arg\max_\pi \mathbb{E}_{(s,a,s')\sim\rho_T^\pi}\left[\log r(s,a) - \alpha\log\left(\rho_T^\pi/\rho_{\widehat{T}}^{\widehat{\pi}}\right)\right] - \alpha D_f\left(\rho_{\widehat{T}}^\pi\|\rho_{\widehat{T}}^{\widehat{\pi}}\right), \tag{7}$$

$$\text{s.t.} \quad \rho^\pi(s,a) = (1-\gamma)\mu_0(s)\pi(a|s) + \gamma\mathcal{T}_\star^\pi\rho^\pi(s,a), \quad \forall(s,a)\in\mathcal{S}\times\mathcal{A}. \tag{8}$$

The challenge of solving this problem is threefold, 1) how to compute the distribution discrepancy term $\log\left(\rho_T^\pi/\rho_{\widehat{T}}^\pi\right)$, 2) how to handle the constraint tractably; 3) how to deal with the unknown distribution $\rho_{\widehat{T}}^\pi(s,a,s')$. To address these, our solution involves three steps:

**Step 1: Computing the distribution discrepancy term.** We denote $R(s,a,s') = \log\left(\rho_T^\pi/\rho_{\widehat{T}}^{\widehat{\pi}}\right)$ for simplicity. Given a tuple $(s,a,s')$, $R(s,a,s')$ characterizes whether it stems from on-policy sampling $\rho_T^\pi(s,a,s')$ or the replay buffer data $\rho_{\widehat{T}}^{\widehat{\pi}}(s,a,s')$. In view of this, we adopt $\mathcal{D}_L$ as a local buffer to collect a small amount of on-policy samples, while $\mathcal{D}_G$ as a global buffer for historical data involving policy and dynamics shifts. Using the notion of GAN (Goodfellow et al., 2014), we can train a discriminator $h(s,a,s')$ to distinguish the tuple $(s,a,s')$ sampled from $\mathcal{D}_L$ or $\mathcal{D}_G$,

$$h^* = \arg\min_h \frac{1}{|\mathcal{D}_G|}\sum_{(s,a,s')\sim\mathcal{D}_G}[\log h(s,a,s')] + \frac{1}{|\mathcal{D}_L|}\sum_{(s,a,s')\sim\mathcal{D}_L}[\log(1-h(s,a,s'))], \tag{9}$$

then the optimal discriminator is solved as $h^*(s,a,s') = \frac{\rho_{\widehat{T}}^{\widehat{\pi}}(s,a,s')}{\rho_{\widehat{T}}^{\widehat{\pi}}(s,a,s')+\rho_T^\pi(s,a,s')}$. Thus, based on the optimal discriminator, we can recover the distribution discrepancies $R(s,a,s')$ by

$$R(s,a,s') = \log\left(\rho_T^\pi(s,a,s')/\rho_{\widehat{T}}^{\widehat{\pi}}(s,a,s')\right) = -\log\left[1/h^*(s,a,s') - 1\right]. \tag{10}$$

**Step 2: Handling the Bellman flow constraint.** In this step, we make a mild assumption that there exists at least one pair of $(s,a)$ to satisfy the constraint (6), ensuring that the constrained optimization problem is feasible. Note that the primal problem (7) is convex, under the feasible assumption, we have that Slater's condition (Boyd & Vandenberghe, 2004) holds. That means, by strong duality, we can adopt $Q(s,a)$ as the Lagrangian multipliers, and the primal problem can be converted to the following equivalent unconstrained problem.

**Proposition 4.2.** *The constraint optimization problem can be transformed into the following unconstrained min-max optimization problem,*

$$\max_\pi \min_{Q(s,a)} (1-\gamma)\mathbb{E}_{s\sim\mu_0,a\sim\pi}[Q(s,a)] - \alpha D_f\left(\rho_{\widehat{T}}^\pi(s,a,s')\|\rho_{\widehat{T}}^{\widehat{\pi}}(s,a,s')\right)$$

$$+ \mathbb{E}_{(s,a,s')\sim\rho_T^\pi}\left[\log r(s,a) - \alpha R(s,a,s') + \gamma\mathcal{T}^\pi Q(s,a) - Q(s,a)\right]. \tag{11}$$

The proof is provided in Appendix A.2.

**Step 3: Optimizing with the data from replay buffer.** To address the issue of the unknown distribution $\rho_{\widehat{T}}^{\pi}(s, a, s')$ in the expectation term, we follow a similar treatment used in the DICE-based methods (Nachum et al., 2019a;b; Nachum & Dai, 2020) and adopt Fenchel conjugate (Fenchel, 2014) to transform the problem (11) into a tractable form, as shown in the following proposition.

**Proposition 4.3.** *Given the accessible distribution $\rho_{\widehat{T}}^{\widehat{\pi}}(s, a, s')$ specified in the global replay buffer, the min-max problem (11) can be transformed as*

$$
\max_{\pi} \min_{Q(s,a)} (1 - \gamma)\mathbb{E}_{s\sim\mu_0, a\sim\pi}[Q(s,a)]
$$
$$
+ \alpha\mathbb{E}_{(s,a,s')\sim\rho_{\widehat{T}}^{\widehat{\pi}}}\left[ f_\star\left( \frac{\log r(s,a) - \alpha R(s,a,s') + \gamma\mathcal{T}^{\pi}Q(s,a) - Q(s,a)}{\alpha} \right) \right], \quad (12)
$$

*where $f_\star(x) := \max_y \langle x, y \rangle - f(y)$ is the Fenchel conjugate of $f$.*

See the proof in Appendix A.3. Such a *min-max* optimization problem allows us to train the policy by randomly sampling in the global replay buffer. Importantly, our method doesn't assume any specific conditions regarding the policy or dynamics shifts in the replay buffer, making it applicable to diverse types of shifts.

## 4.3 PRACTICAL IMPLEMENTATION

Building upon the derived framework, we now introduce a practical learning algorithm called Occupancy-Matching Policy Optimization (OMPO). Apart from the discriminator training, implementing OMPO mainly involves two key designs. More implementation details are in Appendix C.

**Policy learning via bi-level optimization.** For the *min-max* problem (12), we utilize a stochastic first-order two-timescale optimization technique (Borkar, 1997) to iteratively solve the inner objective *w.r.t.* $Q(s,a)$ and the outer one *w.r.t.* $\pi(a|s)$. Such an approach could be regarded as an actor-critic paradigm.

**Instantiations in three settings.** OMPO can be seamlessly instantiated for different policy or dynamics shift settings, one only needs to specify the corresponding interaction data collection scheme. The local buffer $\mathcal{D}_L$ is used to store the fresh data sampled by current policy under the target/desired dynamics; while the global buffer $\mathcal{D}_G$ stores the historical data involving policy and dynamics shifts.

## 5 EXPERIMENT

Our experimental evaluation aims to investigate the following questions: 1) Is OMPO effective in handling the aforementioned three settings with various shifts types? 2) Is the performance consistent with our theoretical analyses?

### 5.1 EXPERIMENTAL RESULTS IN THREE SHIFTED TYPES

Our experiments encompass three distinct scenarios involving policy and dynamics shifts. For each scenario, we employ four popular OpenAI gym benchmarks (Brockman et al., 2016) and their variants, including Hopper-v3, Walker2d-v3, Ant-v3, and Humanoid-v3. Note that, all of experiments involve policy shifts. Since OMPO as well as most baselines are off-policy algorithms, the training data sampled from the replay buffer showcase gaps with the current on-policy distribution.

**Stationary environments.** We conduct a comparison of OMPO with several off-policy model-free baselines by stationary environments. These baselines include: 1) SAC (Haarnoja et al., 2018), the most popular off-policy actor-critic method; 2) TD3 (Fujimoto et al., 2018), which introduces the Double Q-learning technique to mitigate training instability; and 3) AlgaeDICE (Nachum et al., 2019b), utilizing off-policy evaluation methods to reconcile policy gradients to deal with behavior-agnostic and off-policy data. We evaluated all methods using standard benchmarks with stationary dynamics. All methods are trained within the off-policy paradigm.

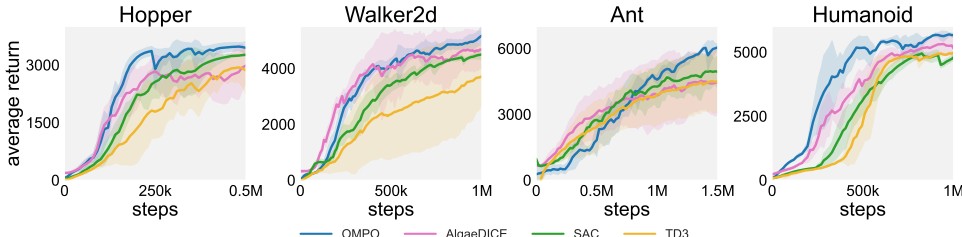

Figure 2: Comparison of learning performance on stationary environments. Solid curves indicate the average performance among five trials under different random seeds, while the shade corresponds to the standard deviation over these trials. We use the same setup for the performance curves below.

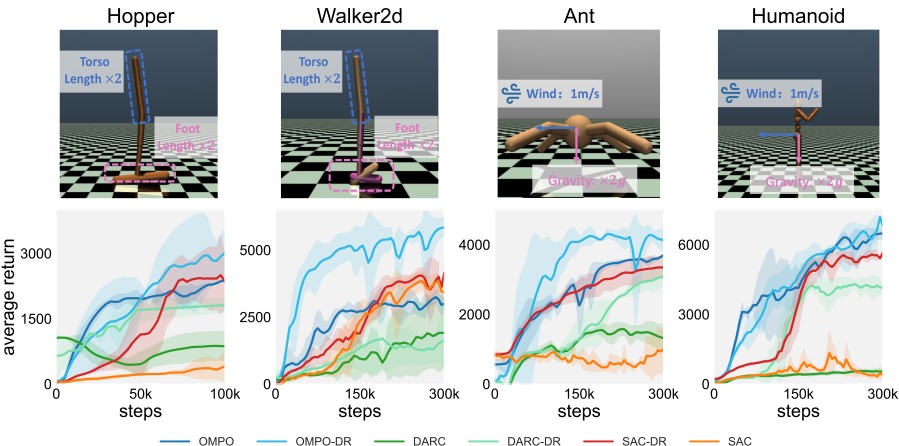

Figure 3: Target dynamics visualizations for the four tasks are on the top. A comparison of learning performance on domain adaption is below. The $x$ coordinate indicates the interaction steps on the target dynamics.

Figure 2 displays the learning curves of the three baselines, along with their asymptotic performance. These results demonstrate OMPO's superior performance in terms of exploration efficiency and training stability, indicating its effectiveness in handling the policy-shifted scenarios.

**Domain adaption.** In this scenario, akin to Eysenbach et al. (2021), the policy trains on both source dynamics ($\widehat{T}$) and target dynamics ($T$). Its objective is to maximize returns efficiently within the target dynamics while collecting ample data from the diverse source dynamics. Across the four tasks, source dynamics align with standard benchmarks, while the target dynamics feature substantial differences. Specifically, in the Hopper and Walker2d tasks, the torso and foot sizes double, and in the Ant and Humanoid tasks, gravity doubles while introducing a headwind with a velocity of $1m/s$. Refer to the top part of Figure 3 for further details.

We benchmark OMPO in this scenario against several baselines, including 1) DARC (Eysenbach et al., 2021), which adjusts rewards for estimating dynamics gaps; 2) Domain Randomization (DR) (Tobin et al., 2017), a technique that randomizes source dynamics parameters to enhance policy adaptability under target dynamics; 3) SAC (Haarnoja et al., 2018), which is directly trained using mixed data from both dynamics. Furthermore, since the DR approach is to randomize the source parameters, DR can be combined with OMPO, DARC and SAC, leading to variants OMPO-DR, DARC-DR and SAC-DR, which provide a comprehensive validation and comparison.

Figure 3 presents the learning curves for all the compared methods, illustrating that OMPO outperforms all baselines with superior eventual performance and high sample efficiency. Notably, when OMPO is combined with DR technology, diverse samplings from randomized source dynamics further harness OMPO's strengths, enabling OMPO-DR to achieve exceptional performance and highlighting its potential for real-world applications. For instance, within the target dynamics of the Walker2d task, OMPO nearly reaches convergence with about 60 trajectories, equivalent to 60,000 steps. More trajectory visualizations are provided in Figure 12, Appendix E.

**Non-stationary environments.** In non-stationary environments, the dynamics vary throughout the training process, setting this scenario apart from domain adaptation scenarios with fixed target dynamics. For the Hopper and Walker2d tasks, the lengths of the torso and foot vary between $0.5-2$

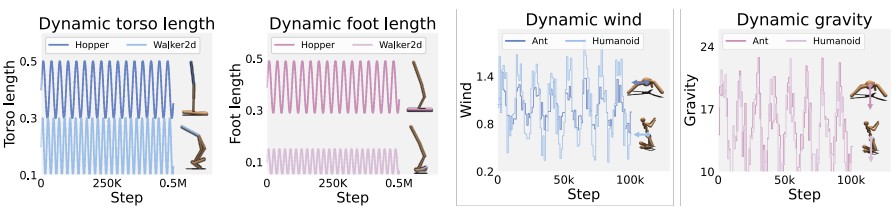

Figure 4: Non-stationarity in structure.    Figure 5: Non-stationarity in mechanics.

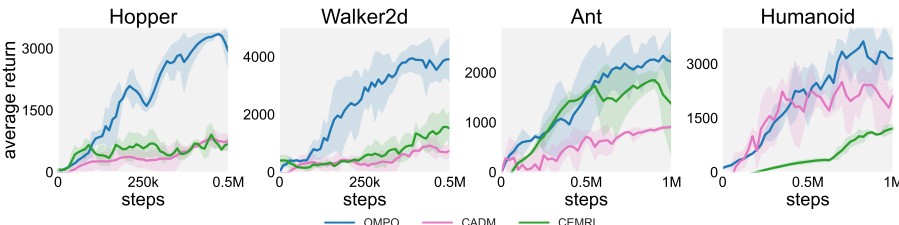

Figure 6: Comparison of learning performance on non-stationary environments.

times the original length. While the Ant and Humanoid tasks feature stochastic variations in gravity $(0.5 - 2 \text{ times})$ and headwinds $(0 - 1.5 m/s)$ at each time step. The non-stationarities of four tasks are depicted in Figures 4 and 5 and comprehensive environment details are provided in Appendix E. The baselines employed in this scenario include: 1) CEMRL (Bing et al., 2022a), which leverages Gaussian mixture models to infer dynamics change; and 2) CaDM Lee et al. (2020), which learns a global dynamics model to generalize across different dynamics.

The results displayed in Figure 6 demonstrate OMPO's ability to effectively handle both policy and dynamics shifts, showing its superiority compared to the baselines. The rapid convergence and automatic data identification of OMPO enable it to adapt seamlessly to diverse shifts, showcasing impressive convergence performance. Besides, even under various non-stationary conditions, our method keeps the same parameters, a notable advantage when compared to the baselines (see hyperparameters and baseline settings in Appendix C).

## 5.2 ANALYSIS OF OMPO UNDER POLICY AND DYNAMICS SHIFTS

**The necessity of handling policy and dynamics shifts.** We visualize the transition occupancy distribution $\rho_T^\pi(s, a, s')$ at different training stages using the training data from the Hopper task within OMPO. As shown in the left part of Figure 7, even under stationary dynamics, policy shifts resulting from constantly updated policies lead to variations of action distributions, thus, $\rho_T^{\pi_1} \neq \rho_T^{\pi_2} \neq \rho_T^{\pi_3}$. When encountering dynamics shifts caused by domain adaptation, as depicted in the right part of Figure 7, these distribution inconsistencies are exacerbated by the dynamics gaps, as evidenced by the differences between $\rho_T^{\pi_1}$ and $\rho_{\widehat{T}}^{\pi_1}$, or $\rho_T^{\pi_2}$ and $\rho_{\widehat{T}}^{\pi_2}$. Furthermore, visualizations of non-stationary environments are provided in Figure 13 of Appendix F, which represent a more complex combination of policy and dynamic shifts.

To further understand the necessity of addressing these shifts, we introduce a variant of OMPO where the distribution discriminator $h(s, a, s')$ is eliminated, disabling the treatment of shifts by setting $R(s, a, s') \equiv 0$. Performance comparisons are shown in Figure 8 using the Hopper task. The results illustrate that in stationary environments, the variant performs comparably to SAC, both of which ignore policy shifts and are weaker than OMPO. Furthermore, when applied in domain adaptation with significant dynamics gaps, the variant suffers from high learning variance and becomes trapped in a local landscape. Similar results appear for non-stationary environments in Figure 14 of Appendix F. These results highlight the effectiveness of our design, as well as the necessity of handling the shifts.

**Ablations on different hyperparameters.** We conducted investigations on two key hyperparameters by Hopper task under non-stationary environments: the size of the local replay buffer $|\mathcal{D}_L|$ and the weighted factor $\alpha$. As shown in Figure 9, our results reveal that choosing a smaller $|\mathcal{D}_L|$ can better capture the policy and dynamics shifts, but it causes training instability of the discriminator, resulting in unstable performance. Conversely, selecting a larger $|\mathcal{D}_L|$ disrupts the freshness of on-policy sampling, resulting in a local landscape.

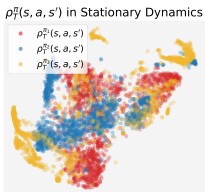
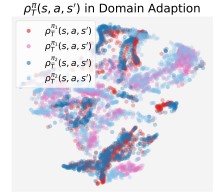
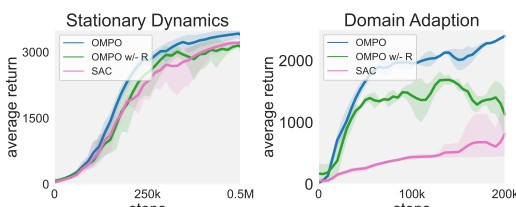

Figure 7: Different stages of $\rho_T^\pi$ by Hopper tasks. Left: 10k, 20k and 50k ($\pi_1$, $\pi_2$ and $\pi_3$). Right: 20k ($\pi_1, T$ and $\pi_1, \widehat{T}$) and 50k ($\pi_2, T$ and $\pi_2, \widehat{T}$).

Figure 8: Performance comparison of OMPO and the variant of OMPO without discriminator by Hopper tasks.

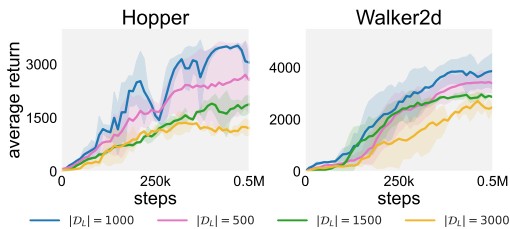
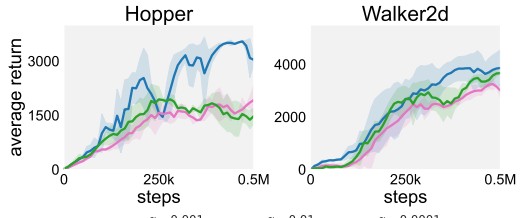

Figure 9: Ablations on the local buffer size $|\mathcal{D}_L|$.

Figure 10: Ablations on weighted factor $\alpha$.

Regarding the weighted factor $\alpha$, as shown in Figure 10, we find that excessively large $\alpha$ makes the $R(s, a, s')$ term overweighted and subordinates the environmental reward during policy optimization. Conversely, excessively small $\alpha$ weakens the discriminator's effectiveness, similar to the issues observed in the OMPO variant without handling shifts.

**Robustness in stochastic robot manipulations.** To further verify OMPO's performance in stochastic robot manipulations, we employ 2 stochastic Panda robot tasks with both dense and sparse rewards (Gallouédec et al., 2021), where random noise is introduced into the actions, and 8 manipulation tasks from Meta-World (Yu et al., 2019) with different objectives (see Appendix E for settings). Table 1 demonstrates that OMPO shows comparable success rates in stochastic environments and outperforms baselines in terms of manipulation tasks. More performance comparisons are provided in Appendix F.3.

Table 1: Success rates of stochastic tasks.

| Tasks | SAC | TD3 | OMPO |
|---|---|---|---|
| Panda-Reach-Den | 92.6% | 94.2% | 97.5% |
| Panda-Reach-Spr | 94.5% | 88.6% | 93.1% |
| Coffer-Push | 15.8% | 3.3% | 68.5% |
| Drawer-Open | 57.7% | 64.3% | 93.4% |
| Door-Unlock | 93.5% | 95.7% | 98.9% |
| Door-Open | 97.5% | 47.9% | 99.5% |
| Hammer | 15.4% | 12.2% | 84.2% |

## 6 CONCLUSION

In this paper, we conduct a holistic investigation of online policy optimization under policy or dynamics shifts. We develop a unified framework to tackle diverse shift settings by introducing a surrogate policy learning objective from the view of transition occupancy matching. Through dual reformulation, we obtain a tractable *min-max* optimization problem, and the practical algorithm OMPO stems from these theoretical analyses. OMPO exhibits superior performance across diverse policy and dynamics shift settings, including policy shifts with stationary environments, domain adaptation, and non-stationary environments, and shows robustness in various challenging locomotion and manipulation tasks. OMPO offers an appealing paradigm for addressing policy and dynamics shifts in many practical RL applications. For example, our empirical results show that OMPO can greatly enhance policy adaptation performance when combined with domain randomization, which can be particularly useful for many sim-to-real transfer problems. Nonetheless, several challenges, such as determining a proper local buffer size to capture the varying on-policy distribution and relaxing the assumption of strictly positive rewards, warrant further investigation. Future work could also extend our work to areas like offline-to-online RL (Li et al., 2023), leveraging simulators with dynamics gaps to enhance offline policy learning (Niu et al., 2022), or hierarchical RL with non-stationarity issues in high-level policy optimization (Nachum et al., 2018).

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

# A    Proofs in the Main Text

Here, we first present a sketch of theoretical analyses in Figure 11. We first propose a surrogate objective to handle policy and dynamics shifts (Equation 5). Then, to make this objective tractable, we consider the Bellman flow constraint (Equation 6) thus constructing a constraint optimization problem (Equations 7 and 8). To solve this problem, we divide it into three steps. (1) We deal with the distribution discrepancy $R(s, a, s')$ by the discriminator $h^*(s, a, s')$ (Equation 10); (2) We handle the Bellman flow constraint by Lagrangian relaxation (Equation 11); (3) To get rid of the unknown distribution $\rho_{\widehat{T}}^\pi$, we utilize Fenchel conjugate to obtain the final tractable optimization problem (Equation 12).

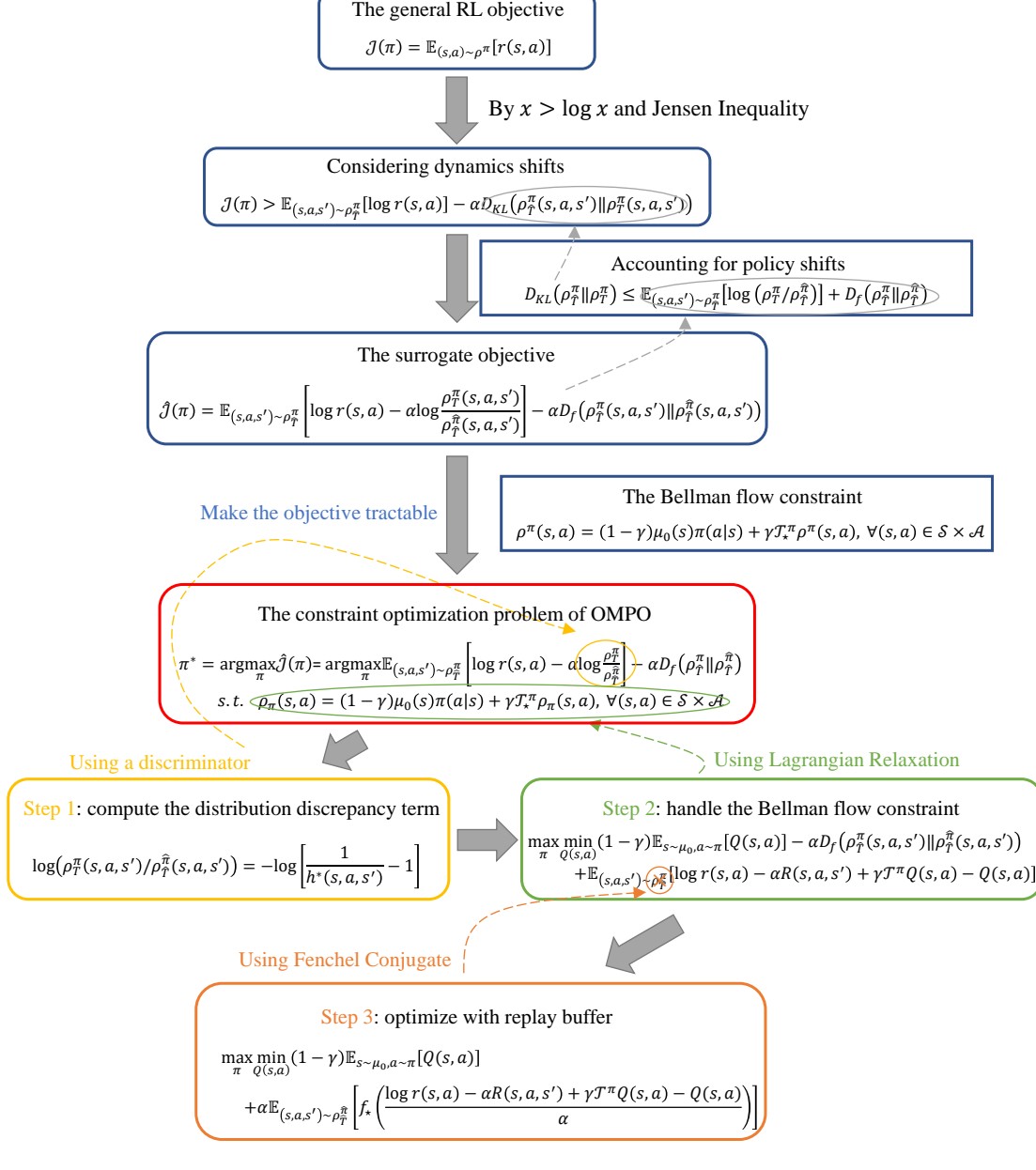

Figure 11: Theoretical sketch of OMPO.

## A.1 PROOF OF PROPOSITION 4.1

**Proposition A.1.** *Let $\rho_{\widehat{T}}^{\pi}(s, a, s')$ denote the transition occupancy distribution specified by the replay buffer. The following inequality holds for any $f$-divergence that upper bounds the KL divergence:*

$$D_{\mathrm{KL}}\left(\rho_{\widehat{T}}^{\pi}\|\rho_T^{\pi}\right) \le \mathbb{E}_{(s,a,s')\sim\rho_{\widehat{T}}^{\pi}}\left[\log\left(\rho_T^{\pi}/\rho_{\widehat{T}}^{\widehat{\pi}}\right)\right] + D_f\left(\rho_{\widehat{T}}^{\pi}\|\rho_{\widehat{T}}^{\widehat{\pi}}\right). \tag{13}$$

*Proof.* Based on the definition of KL-divergence, we have

$$
\begin{aligned}
D_{\mathrm{KL}}\left(\rho_{\widehat{T}}^{\pi}(s,a,s')\|\rho_T^{\pi}(s,a,s')\right) &= \mathbb{E}_{(s,a,s')\sim\rho_{\widehat{T}}^{\pi}}\left[\log\frac{\rho_T^{\pi}(s,a,s')}{\rho_{\widehat{T}}^{\pi}(s,a,s')}\right] \\
&= \mathbb{E}_{(s,a,s')\sim\rho_{\widehat{T}}^{\pi}}\left[\log\left(\frac{\rho_T^{\pi}(s,a,s')}{\rho_{\widehat{T}}^{\widehat{\pi}}(s,a,s')} \cdot \frac{\rho_{\widehat{T}}^{\widehat{\pi}}(s,a,s')}{\rho_{\widehat{T}}^{\pi}(s,a,s')}\right)\right] \\
&= \mathbb{E}_{(s,a,s')\sim\rho_{\widehat{T}}^{\pi}}\left[\log\left(\frac{\rho_T^{\pi}(s,a,s')}{\rho_{\widehat{T}}^{\widehat{\pi}}(s,a,s')}\right)\right] + \mathbb{E}_{(s,a,s')\sim\rho_{\widehat{T}}^{\pi}}\left[\log\left(\frac{\rho_{\widehat{T}}^{\widehat{\pi}}(s,a,s')}{\rho_{\widehat{T}}^{\pi}(s,a,s')}\right)\right] \\
&= \mathbb{E}_{(s,a,s')\sim\rho_{\widehat{T}}^{\pi}}\left[\log\left(\frac{\rho_T^{\pi}(s,a,s')}{\rho_{\widehat{T}}^{\widehat{\pi}}(s,a,s')}\right)\right] + D_{\mathrm{KL}}\left(\rho_{\widehat{T}}^{\pi}(s,a,s')\|\rho_{\widehat{T}}^{\widehat{\pi}}(s,a,s')\right) \\
&\le \mathbb{E}_{(s,a,s')\sim\rho_{\widehat{T}}^{\pi}}\left[\log\left(\frac{\rho_T^{\pi}(s,a,s')}{\rho_{\widehat{T}}^{\widehat{\pi}}(s,a,s')}\right)\right] + D_f\left(\rho_{\widehat{T}}^{\pi}(s,a,s')\|\rho_{\widehat{T}}^{\widehat{\pi}}(s,a,s')\right). \quad \text{by any} \quad D_f \ge D_{\mathrm{KL}}
\end{aligned}
\tag{14}
$$

The proof is completed. $\square$

## A.2 PROOF OF PROPOSITION 4.2

**Proposition A.2.** *The constraint optimization problem can be transformed into an unconstrained min-max problem,*

$$\max_{\pi}\min_{Q(s,a)}(1-\gamma)\mathbb{E}_{s\sim\mu_0,a\sim\pi}[Q(s,a)] - \alpha D_f\left(\rho_{\widehat{T}}^{\pi}(s,a,s')\|\rho_{\widehat{T}}^{\widehat{\pi}}(s,a,s')\right)$$
$$+ \mathbb{E}_{s,a,s'\sim\rho_{\widehat{T}}^{\pi}}\left[\log r(s,a) - \alpha R(s,a,s') + \gamma\mathcal{T}^{\pi}Q(s,a) - Q(s,a)\right]. \tag{15}$$

*Proof.* With the primal optimization,

$$\pi^* = \arg\max_{\pi}\widehat{\mathcal{J}}(\pi) = \arg\max_{\pi}\mathbb{E}_{(s,a,s')\sim\rho_{\widehat{T}}^{\pi}}\left[\log r(s,a) - \alpha\log\left(\rho_T^{\pi}/\rho_{\widehat{T}}^{\widehat{\pi}}\right)\right] - \alpha D_f\left(\rho_{\widehat{T}}^{\pi}\|\rho_{\widehat{T}}^{\widehat{\pi}}\right),$$
$$\text{s.t.} \quad \rho^{\pi}(s,a) = (1-\gamma)\mu_0(s)\pi(a|s) + \gamma\mathcal{T}_{\star}^{\pi}\rho^{\pi}(s,a), \quad \forall(s,a)\in\mathcal{S}\times\mathcal{A},$$

let $Q(s, a)$ denote the Lagrangian multipliers, then we have

$$\mathbb{E}_{(s,a,s')\sim\rho_{\widehat{T}}^{\pi}}\left[\log r(s,a) - \alpha R(s,a,s')\right] - \alpha D_f\left(\rho_{\widehat{T}}^{\pi}(s,a,s')\|\rho_{\widehat{T}}^{\widehat{\pi}}(s,a,s')\right)$$
$$+ \sum_{s,a} Q(s,a)\left[(1-\gamma)\mu_0(s)\pi(a|s) + \gamma\mathcal{T}_\star^\pi\rho^\pi(s,a) - \rho^\pi(s,a)\right]$$

$$= \mathbb{E}_{(s,a,s')\sim\rho_{\widehat{T}}^{\pi}}\left[\log r(s,a) - \alpha R(s,a,s')\right] - \alpha D_f\left(\rho_{\widehat{T}}^{\pi}(s,a,s')\|\rho_{\widehat{T}}^{\widehat{\pi}}(s,a,s')\right)$$
$$+ \sum_{s,a} Q(s,a)\left[(1-\gamma)\mu_0(s)\pi(a|s) + \gamma\pi(a|s)\sum_{\hat{s},\hat{a}}\rho_{\widehat{T}}^{\pi}(\hat{s},\hat{a},s) - \sum_{s'}\rho_{\widehat{T}}^{\pi}(s,a,s')\right]$$

$$= \mathbb{E}_{(s,a,s')\sim\rho_{\widehat{T}}^{\pi}}\left[\log r(s,a) - \alpha R(s,a,s')\right] - \alpha D_f\left(\rho_{\widehat{T}}^{\pi}(s,a,s')\|\rho_{\widehat{T}}^{\widehat{\pi}}(s,a,s')\right)$$
$$+ (1-\gamma)\sum_{s,a} Q(s,a)\pi(a|s)\mu_0(s) + \gamma\sum_{s,a} Q(s,a)\pi(a|s)\sum_{\hat{s},\hat{a}}\rho_{\widehat{T}}^{\pi}(\hat{s},\hat{a},s)$$
$$- \sum_{s,a,s'}\rho_{\widehat{T}}^{\pi}(s,a,s')Q(s,a)$$

$$= \mathbb{E}_{(s,a,s')\sim\rho_{\widehat{T}}^{\pi}}\left[\log r(s,a) - \alpha R(s,a,s')\right] - \alpha D_f\left(\rho_{\widehat{T}}^{\pi}(s,a,s')\|\rho_{\widehat{T}}^{\widehat{\pi}}(s,a,s')\right)$$
$$+ (1-\gamma)\sum_{s,a} Q(s,a)\pi(a|s)\mu_0(s) + \gamma\sum_{s',a'} Q(s',a')\pi(a'|s')\sum_{s,a}\rho_{\widehat{T}}^{\pi}(s,a,s')$$
$$- \sum_{s,a,s'}\rho_{\widehat{T}}^{\pi}(s,a,s')Q(s,a)$$

$$= \mathbb{E}_{(s,a,s')\sim\rho_{\widehat{T}}^{\pi}}\left[\log r(s,a) - \alpha R(s,a,s')\right] - \alpha D_f\left(\rho_{\widehat{T}}^{\pi}(s,a,s')\|\rho_{\widehat{T}}^{\pi}(s,a,s')\right)$$
$$+ (1-\gamma)\mathbb{E}_{s\sim\mu_0,a\sim\pi}Q(s,a) + \sum_{s,a,s'}\rho_{\widehat{T}}^{\pi}(s,a,s')\left[\gamma\sum_{a'} Q(s',a')\pi(a'|s') - Q(s,a)\right]$$

$$= (1-\gamma)\mathbb{E}_{s\sim\mu_0,a\sim\pi}[Q(s,a)] - \alpha D_f\left(\rho_{\widehat{T}}^{\pi}(s,a,s,')\|\rho_{\widehat{T}}^{\widehat{\pi}}(s,a,s,')\right)$$
$$+ \mathbb{E}_{(s,a,s')\sim\rho_{\widehat{T}}^{\pi}}\left[\log r(s,a) - \alpha R(s,a,s') + \gamma\mathcal{T}^\pi Q(s,a) - Q(s,a)\right]. \tag{16}$$

The proof is completed. $\qquad\square$

## A.3    PROOF OF PROPOSITION 4.3

We first briefly introduce the Fenchel conjugate, which is a crutical technology in the proof of Proposition 4.3.

**Definition A.3** (Fenchel conjugate). *In a real Hilbert space $\mathcal{X}$, if a function $f(x)$ is proper, then the Fenchel conjugate $f_\star$ of $f$ is defined as*

$$f_\star(x) = \sup_{y\in\mathcal{X}}\langle x,y\rangle - f(y), \tag{17}$$

*where the domain of the $f_\star(x)$ is given by:*

$$dom\ f_\star = \left\{x: \sup_{y\in dom\ f}(\langle x,y\rangle - f(y)) < \infty\right\}. \tag{18}$$

Based on this definition, we have $f_{\star\star}(x) = f(x) = \min_{y\in\mathcal{X}} f_\star(y) - \langle x,y\rangle$. For the $f$-divergence function, we let $D_f(x\|p) = \mathbb{E}_{z\sim p}f(x/p)$, thus its Fenchel conjugate is $\mathbb{E}_{z\sim p}[f_\star(y(z))]$ (Fenchel, 2014). Further, we apply this property into the $f$-divergence term $D_f\left(\rho_{\widehat{T}}^{\pi}\|\rho_{\widehat{T}}^{\widehat{\pi}}\right)$, and we have

$$D_f\left(\rho_{\widehat{T}}^{\pi}(s,a,s')\|\rho_{\widehat{T}}^{\widehat{\pi}}(s,a,s')\right) = \min_{y(s,a,s')}\mathbb{E}_{(s,a,s')\sim\rho_{\widehat{T}}^{\widehat{\pi}}}\left[f_\star(y(s,a,s'))\right] - \mathbb{E}_{(s,a,s')\sim\rho_{\widehat{T}}^{\pi}}\left[y(s,a,s')\right].$$
$$\tag{19}$$

With the help of the derivation , we start the proof of Proposition 4.3.

**Proposition A.4.** *Given the accessible distribution $\rho_{\widehat{\mathcal{T}}}^{\widehat{\pi}}(s, a, s')$ specified in the global replay buffer, the min-max problem (11) can be transformed as*

$$\max_{\pi} \min_{Q(s,a)} (1 - \gamma)\mathbb{E}_{s\sim\mu_0, a\sim\pi}[Q(s,a)]$$
$$+ \alpha\mathbb{E}_{(s,a,s')\sim\rho_{\widehat{\mathcal{T}}}^{\widehat{\pi}}} \left[ f_\star \left( \frac{\log r(s,a) - \alpha R(s,a,s') + \gamma\mathcal{T}^\pi Q(s,a) - Q(s,a)}{\alpha} \right) \right], \qquad (20)$$

*where $f_\star(x) := \max_y \langle x, y\rangle - f(y)$ is the Fenchel conjugate of $f$.*

*Proof.* For the proposed *min-max* problem (11), we have

$$\max_{\pi} \min_{Q(s,a)} (1 - \gamma)\mathbb{E}_{s\sim\mu_0, a\sim\pi}[Q(s,a)] - \alpha D_f\left(\rho_{\widehat{\mathcal{T}}}^\pi(s,a,s') \| \rho_{\widehat{\mathcal{T}}}^{\widehat{\pi}}(s,a,s')\right)$$
$$+ \mathbb{E}_{s,a,s'\sim\rho_{\widehat{\mathcal{T}}}^\pi} [\log r(s,a) - \alpha R(s,a,s') + \gamma\mathcal{T}^\pi Q(s,a) - Q(s,a)]$$
$$= \max_{\pi} \min_{Q(s,a)} \mathbb{E}_{s,a,s'\sim\rho_{\widehat{\mathcal{T}}}^\pi} [\log r(s,a) - \alpha R(s,a,s') + \gamma\mathcal{T}^\pi Q(s,a) - Q(s,a)]$$
$$- \alpha D_f\left(\rho_{\widehat{\mathcal{T}}}^\pi(s,a,s') \| \rho_{\widehat{\mathcal{T}}}^{\widehat{\pi}}(s,a,s')\right) + (1-\gamma)\mathbb{E}_{s\sim\mu_0, a\sim\pi}[Q(s,a)]. \qquad (21)$$

Then, for the $f$-divergence term $D_f\left(\rho_{\widehat{\mathcal{T}}}^\pi \| \rho_{\widehat{\mathcal{T}}}^{\widehat{\pi}}\right)$, we apply its Fenchel Conjugate (19) into (21),

$$\max_{\pi} \min_{Q(s,a)} \mathbb{E}_{s,a,s'\sim\rho_{\widehat{\mathcal{T}}}^\pi} [\log r(s,a) - \alpha R(s,a,s') + \gamma\mathcal{T}^\pi Q(s,a) - Q(s,a)]$$
$$- \alpha D_f\left(\rho_{\widehat{\mathcal{T}}}^\pi(s,a,s') \| \rho_{\widehat{\mathcal{T}}}^{\widehat{\pi}}(s,a,s')\right) + (1-\gamma)\mathbb{E}_{s\sim\mu_0, a\sim\pi}[Q(s,a)]$$
$$= \max_{\pi} \min_{Q(s,a)} \min_{y(s,a,s')} \mathbb{E}_{s,a,s'\sim\rho_{\widehat{\mathcal{T}}}^\pi} [\log r(s,a) - \alpha R(s,a,s') + \gamma\mathcal{T}^\pi Q(s,a) - Q(s,a)] \qquad \text{By (19)}$$
$$+ \alpha\mathbb{E}_{(s,a,s')\sim\rho_{\widehat{\mathcal{T}}}^{\widehat{\pi}}} [f_\star(y(s,a,s'))] - \alpha\mathbb{E}_{(s,a,s')\sim\rho_{\widehat{\mathcal{T}}}^\pi} [y(s,a,s')] + (1-\gamma)\mathbb{E}_{s\sim\mu_0, a\sim\pi}[Q(s,a)]$$
$$= \max_{\pi} \min_{Q(s,a)} \min_{y(s,a,s')} \mathbb{E}_{s,a,s'\sim\rho_{\widehat{\mathcal{T}}}^\pi} [\log r(s,a) - \alpha R(s,a,s') + \gamma\mathcal{T}^\pi Q(s,a) - Q(s,a) - \alpha y(s,a,s')]$$
$$+ \alpha\mathbb{E}_{(s,a,s')\sim\rho_{\widehat{\mathcal{T}}}^{\widehat{\pi}}} [f_\star(y(s,a,s'))] + (1-\gamma)\mathbb{E}_{s\sim\mu_0, a\sim\pi}[Q(s,a)]. \qquad (22)$$

To eliminate the expectation over $\rho_{\widehat{\mathcal{T}}}^\pi(s, a, s')$, we follow prior works (Nachum et al., 2019b;a) and make a change of variables by

$$y(s, a, s') = \frac{\log r(s,a) - \alpha R(s,a,s') + \gamma\mathcal{T}^\pi Q(s,a) - Q(s,a)}{\alpha}. \qquad (23)$$

Note that in this variable changing, $\min y(s, a, s')$ can be equivalent to $\min \mathcal{T}^\pi Q(s, a) - Q(s, a)$ (we suppose $\alpha > 0$ and $r(s, a), R(s, a, s')$ are both irrelevant variables). Based on the definition of $\mathcal{T}^\pi$, we find that in the inner optimization problem of (22) with the fixed variable $\pi$, $\min y(s, a, s')$ can be replaced by $\min Q(s, a)$. Thus, we can further yield

$$\max_{\pi} \min_{Q(s,a)} \min_{y(s,a,s')} \mathbb{E}_{s,a,s'\sim\rho_{\widehat{\mathcal{T}}}^\pi} [\log r(s,a) - \alpha R(s,a,s') + \gamma\mathcal{T}^\pi Q(s,a) - Q(s,a) - \alpha y(s,a,s')]$$
$$+ \alpha\mathbb{E}_{(s,a,s')\sim\rho_{\widehat{\mathcal{T}}}^{\widehat{\pi}}} [f_\star(y(s,a,s'))] + (1-\gamma)\mathbb{E}_{s\sim\mu_0, a\sim\pi}[Q(s,a)]$$
$$= \max_{\pi} \min_{Q(s,a)} (1-\gamma)\mathbb{E}_{s\sim\mu_0, a\sim\pi}[Q(s,a)]$$
$$+ \alpha\mathbb{E}_{(s,a,s')\sim\rho_{\widehat{\mathcal{T}}}^{\widehat{\pi}}} \left[ f_\star \left( \frac{\log r(s,a) - \alpha R(s,a,s') + \gamma\mathcal{T}^\pi Q(s,a) - Q(s,a)}{\alpha} \right) \right]. \qquad (24)$$

The proof is completed. $\qquad\square$

# B    More Discussion of Performance Improvemeng for OMPO

Here, we discuss the performance improvement of OMPO from an optimization objective perspective, compared to the general RL objective (1). To recap, our proposed surrogate objective to handle policy and dynamics shifts is formulated as

$$\widehat{\mathcal{J}}(\pi) = \mathbb{E}_{(s,a,s')\sim\rho_{\widehat{T}}^{\pi}} \left[ \log r(s,a) - \alpha \log \frac{\rho_T^{\pi}(s,a,s')}{\rho_{\widehat{T}}^{\widehat{\pi}}(s,a,s')} \right] - \alpha D_f \left( \rho_{\widehat{T}}^{\pi}(s,a,s') \| \rho_{\widehat{T}}^{\widehat{\pi}}(s,a,s') \right).$$

**Ideal Condition without Policy and Dynamics Shifts.**    When there is no policy and dynamics shifts, we have $\widehat{T} = T$ and $\widehat{\pi} = \pi$. Thus, the negative terms in the surrogate objective satisfy

$$\log \frac{\rho_T^{\pi}(s,a,s')}{\rho_{\widehat{T}}^{\widehat{\pi}}(s,a,s')} = \log \frac{\rho_T^{\pi}(s,a,s')}{\rho_T^{\pi}(s,a,s')} = 0,$$

$$D_f \left( \rho_{\widehat{T}}^{\pi}(s,a,s') \| \rho_{\widehat{T}}^{\widehat{\pi}}(s,a,s') \right) = D_f \left( \rho_T^{\pi}(s,a,s') \| \rho_T^{\pi}(s,a,s') \right) = 0.$$

At this case, the surrogate objective reduced to

$$\widehat{\mathcal{J}}(\pi) = \mathbb{E}_{(s,a,s')\sim\rho_{\widehat{T}}^{\pi}} \left[ \log r(s,a) \right]. \tag{25}$$

This actually corresponds to solving an MDP with reward shaping using the logarithmic function. Since the logarithmic function is monotonically increasing, it does not largely change the nature of the original task.

**Only Policy shifts without Dynamics shifts.**    In stationary environments where $\widehat{T} = T$, the training data exhibit policy shifts since they are collected by various policy in the training. The general RL objective (1),

$$\pi^* = \arg\max_{\pi} \mathbb{E}_{(s,a)\sim\rho^{\pi}} [r(s,a)],$$

assumes the distribution from on-policy samplings $(s,a) \sim \rho^{\pi}$. Under policy shifts, the training data $(s,a) \sim \rho^{\widehat{\pi}}$ have a mismatch to on-policy samplings $(s,a) \sim \rho^{\pi}$, resulting in suboptimal performance. While, for OMPO, the surrogate objective (5) reduces to

$$\begin{aligned}
\widehat{\mathcal{J}}(\pi) &= \mathbb{E}_{(s,a,s')\sim\rho_T^{\pi}} \left[ \log r(s,a) - \alpha \log(\rho_T^{\pi}/\rho_T^{\widehat{\pi}}) \right] - \alpha D_f \left( \rho_T^{\pi} \| \rho_T^{\widehat{\pi}} \right) \\
&= \mathbb{E}_{(s,a)\sim\rho^{\pi}} \left[ \log r(s,a) - \alpha \log \left( \rho^{\pi}/\rho^{\widehat{\pi}} \right) \right] - \alpha D_f \left( \rho^{\pi} \| \rho^{\widehat{\pi}} \right) \\
&= \mathbb{E}_{(s,a)\sim\rho^{\pi}} [\log r(s,a)] - \alpha \left[ D_{KL} \left( \rho^{\pi} \| \rho^{\widehat{\pi}} \right) + D_f \left( \rho^{\pi} \| \rho^{\widehat{\pi}} \right) \right], \tag{26}
\end{aligned}$$

which essentially regularizes the discrepancy between on-policy occupancy $\rho^{\pi}$ and the occupancy induced by off-policy samples $\rho^{\widehat{\pi}}$ from the replay buffer, which helps to alleviate potential instability caused by off-policy learning (Liu et al., 2019; Xue et al., 2023).

**Policy shifts with Dynamics Shifts.**    For the most general setting, Section 4.2 has discussed the solution. By carefully handling the impact of polic and dynamics shifts, OMPO can achieve better performance than the general RL objective in practice.

## C    IMPLEMENTATION DETAILS

In this section, we delve into the specific implementation details of OMPO. To do so, we employ deep neural networks parameterized by $\phi$, $\theta$, and $\psi$ to represent the discriminator $h(s, a, s')$, critic $Q(s, a)$, and policy $\pi(a|s)$, respectively. Here are the key aspects of the implementation:

**Discriminator training.**    To address practical considerations during online training, it's essential to manage the discrepancy in data volume between the local buffer $\mathcal{D}_L$ and the global buffer $\mathcal{D}_G$. To tackle this challenge, we adopt the following strategy: At each gradient step, we randomly draw several batches, each with a size of $|\mathcal{D}_L|$, and employ them to train the discriminator. This process ensures a balanced use of data from both $\mathcal{D}_L$ and $\mathcal{D}_G$ during training.

**Specialized actor-critic architecture.**    For the $f$-divergence, we specifically choose $f(x) = \frac{1}{p}(x - 1)^p$, with its Fenchel conjugate denoted as $f_\star(x) = \frac{1}{q}x^q + x$, where $\frac{1}{p} + \frac{1}{q} = 1$. To practically address the tractable *min-max* optimization problem (12), we initially solve the inner problem concerning $Q(s, a)$ using a gradient-based approach:

$$Q(s, a) \leftarrow \arg\min_Q (1 - \gamma)\mathbb{E}_{s \sim \mu_0, a \sim \pi}[Q(s, a)]$$

$$+ \alpha\mathbb{E}_{(s,a,s') \sim \rho_{\widehat{T}}^{\widehat{\pi}}}\left[ f_\star\left( \frac{\log r(s, a) - \alpha R(s, a, s') + \gamma \mathcal{T}^\pi Q(s, a) - Q(s, a)}{\alpha} \right) \right]. \qquad (27)$$

Subsequently, employing the policy gradient method (Sutton et al., 1999; Nachum et al., 2019b), where:

$$\frac{\partial}{\partial\pi} \min_Q J(\pi, Q) = \mathbb{E}_{(s,a) \sim \rho_\pi}\left[ \tilde{Q}(s, a)\nabla \log \pi(a|s) \right], \qquad (28)$$

with $\tilde{Q}(s, a)$ representing the Q-value function of $\pi$ based on rewards $\tilde{r}(s, a) = r(s, a) - \alpha f'(\rho_T^\pi / \rho_{\widehat{T}}^{\widehat{\pi}})$, updated using $\tilde{Q}(s, a)$ from the inner problem, we update the policy $\pi$ as follows:

$$\pi(a|s) \leftarrow \arg\min_\pi (1 - \gamma)\mathbb{E}_{s \sim \mu_0, a \sim \pi}[\tilde{Q}(s, a)]$$

$$+ \alpha\mathbb{E}_{(s,a,s') \sim \rho_{\widehat{T}}^{\widehat{\pi}}}\left[ f'_\star\left( \frac{\log r(s, a) - \alpha R(s, a, s') + \gamma \mathcal{T}^\pi \tilde{Q}(s, a) - \tilde{Q}(s, a)}{\alpha} \right) \right]. \qquad (29)$$

Here, $f'_\star(x) = x^{q-1} + 1$ represents the derivative of $f_\star(x)$.

In the actor-critic architecture, we follow a two-step process: first, we update the critic network, and then we update the actor network. To ensure training stability, we implement a stochastic first-order two-time scale optimization technique (Borkar, 1997), where the gradient update step size for the inner problem is significantly larger than that for the outer layer. This setup ensures rapid convergence of the inner problem to suit the outer problem.

Especially, to deal with $s_0 \sim \mu_0$ in Equation (27) and Equation (29), we refer to the implementation of previous DICE works (Nachum et al., 2019b; Ma et al., 2022), and adopt an initial-state buffer to store initial state $s_0$. When optimizing Equation (27) and Equation (29), we can sample $s_0$ from the initial-state buffer.

**Application in different scenarios.**    With the proposed actor-critic architecture, OMPO seamlessly accommodates various scenarios involving diverse shifts. By employing the local replay buffer $\mathcal{D}_L$ to collect fresh data and the global replay buffer $\mathcal{D}_G$ to store all historical data, OMPO effectively addresses different shift scenarios:

- *Policy shifts with stationary dynamics*: In this scenario, only the fresh data is retained in $\mathcal{D}_L$. When $\mathcal{D}_L$ reaches its capacity, we initiate training of the discriminator. Subsequently, we sample random batches from $\mathcal{D}_G$ to update both the critic and the actor, with the updated discriminator. Following this update, we merge the data in $\mathcal{D}_L$ into $\mathcal{D}_G$ and reset $\mathcal{D}_L$ for further data collection.

- *Policy shifts with domain adaption*: When policy shifts involve domain adaptation, fresh data sampled under the target dynamics is stored in $\mathcal{D}_L$, while data from the source dynamics resides in $\mathcal{D}_G$. Then, the training process mirrors that of the scenario with stationary dynamics.

---

**Algorithm 1:** Occupancy-Matching Policy Optimization (OMPO)

---

**initialize:** Global buffer $\mathcal{D}_G$, local buffer $\mathcal{D}_L$, initial-state buffer $\mathcal{D}_0$, critic $Q_\theta$, policy $\pi_\psi$,
discriminator $h$

**repeat**

    **for** *each environment step* **do**

        **if** *Initialisation* **then**

            └ Store the initial state $s_0 \sim \mu_0$ into $\mathcal{D}_0$

        `/* Case 1:  Interact with stationary environment        */`

        Collect $(s, a, s', r)$ with $\pi_\psi$ from environment; add to $\mathcal{D}_L$

        `/* Case 2:  Interact with multiple domains for adaption */`

        Collect $(s, a, s', r)$ with $\pi_\psi$ from source domains; add to $\mathcal{D}_G$

        Collect $(s, a, s', r)$ with $\pi_\psi$ from target domain; add to $\mathcal{D}_L$

        `/* Case 3:  Interact with non-stationary environment    */`

        Collect $(s, a, s', r)$ with $\pi_\psi$ from current environment; add to $\mathcal{D}_L$

    **if** $\mathcal{D}_L$ *is full* **then**

        **for** *each gradient step* **do**

            Update discriminator $h(s, a, s')$ by Eq.(9) from both $\mathcal{D}_G$ and $\mathcal{D}_L$

            Computing $R(s, a, s')$ with discriminator $h(s, a, s')$ by Eq.(10)

            Update critic $Q_\theta$ by Eqs.(27) and actor $\pi_\psi$ by Eqs.(29) from $\mathcal{D}_G$ and $\mathcal{D}_0$

        Merge global buffer by $\mathcal{D}_G \leftarrow \mathcal{D}_G \cup \mathcal{D}_L$ and reset local buffer $\mathcal{D}_L \leftarrow \emptyset$

**until** *the policy performs well in the environment*;

---

- *Policy shifts with non-stationary dynamics*: The training process aligns with the first scenario regardless of policy and dynamics shifts.

Therefore, in various scenarios, adjusting data collection in distinct replay buffers suffices, eliminating the need for any modifications to the policy optimization process. These approaches are succinctly summarized in Algorithm 1.

## C.1   HYPERPARAMETERS AND NETWORK ARCHITECTURE

We use the same hyperparameters for all OMPO experiments in this paper. In terms of architecture, we use a simple 2-layer ReLU network with a hidden size of 256 to parameterize the cirtic network. For the policy network, we use the same architecture to parameterize a Gaussian distribution, where the mean and the log standard deviation are outputs of two separate heads, referring to SAC (Haarnoja et al., 2018). For the discriminator network, we also a simple 2-layer network. Table 2 summarizes the hyperparameters as well as the architecture.

## C.2   BASELINES

In our experiments across the three different scenarios, we have implemented all the baseline algorithms using their original code bases to ensure a fair and consistent comparison.

For the stationary environments,

- For SAC (Haarnoja et al., 2018), we utilized the open-source PyTorch implementation, available at `https://github.com/pranz24/pytorch-soft-actor-critic`.

- TD3 (Fujimoto et al., 2018) was integrated into our experiments through its official codebase, accessible at `https://github.com/sfujim/TD3`.

- AlgaeDICE (Nachum et al., 2019b) was employed with its official implementation from `https://github.com/google-research/google-research/tree/master/algae_dice`.

For the domain adaption,

Table 2: The hyperparameters of OMPO

| | Hyperparameter | Value |
|---|---|---|
| OMPO Hyperparameters | Optimizer | Adam |
| | Critic learning rate | 3e-4 |
| | Actor learning rate | 1e-4 |
| | Discount factor | 0.99 |
| | Mini-batch | 256 |
| | Actor Log Std. Clipping | $(-20, 2)$ |
| | Local buffer size | 1000 |
| | Global buffer size | 1e6 |
| | Order $q$ of Conjugate function | 1.5 |
| | Weighted factor | 0.001 |
| Architecture | Critic hidden dim | 256 |
| | Critic hidden layers | 2 |
| | Critic activation function | elu |
| | Actor hidden dim | 256 |
| | Actor hidden layers | 2 |
| | Actor activation function | elu |
| | Discriminator hidden dim | 256 |
| | Discriminator hidden layers | 2 |
| | Discriminator activation function | tanh |

- DARC (Eysenbach et al., 2021) was harnessed via its official implementation, found at `https://github.com/google-research/google-research/tree/master/darc`.
- We meticulously detailed the implementation of Domain Randomization (Tobin et al., 2017) in Appendix E.

For the non-stationary environments,

- CaDM (Lee et al., 2020) was implemented using its official codebase accessible at `https://github.com/younggyoseo/CaDM`.
- CEMRL (Bing et al., 2022a) was utilized with its official implementation found at `https://github.com/zhenshan-bing/cemrl`.

# D  COMPARISON WITH PRIOR DICE WORKS

In comparison to prior works within the DICE family, OMPO distinguishes itself as primarily tailored for **Online**, **Shifted** scenarios. This distinction is notable in terms of both theoretical underpinnings and generalizability.

- **Generalizability**

  - **Not only policy shifts, but also dynamics shifts**: While numerous DICE works concentrate on discrepancies in state or state-action occupancy distributions, such as DualDICE (Nachum et al., 2019a) and AlgaeDICE (Nachum et al., 2019b), their primary concern is variations in data distributions due to differing policies (*i.e.*, behavior-agnostic and off-policy data distribution in their paper). Consequently, these approaches may struggle when policy shifts and dynamic shifts co-occur, as they do not account for the transition dynamics for the next state when given the current states and actions.

  - **Model-Based Distinctions**: TOM (Ma et al., 2023), as a model-based RL method, also employs transition occupancy distributions. However, a fundamental difference exists between TOM and OMPO. TOM encourages the learned model to consider policy exploration while optimizing the transition occupancy distribution, *i.e.*, $\min_{\hat{T}} D_f(d_{\hat{T}}^\pi(s, a, s') \| d_T^\pi(s, a, s'))$, but it does not incorporate the environmental reward into its objective. In OMPO, our objective seeks to identify similar experiences collected from the global buffer, with a focus on enhancing environmental returns, see the $\log r(s, a)$ term in the surrogate objective (5). Furthermore, TOM can only apply to stationary environments and does not address policy shifts and dynamic shifts explicitly.

  - **Experimental Effectiveness**: Through our experimental results, OMPO demonstrates its efficacy across diverse scenarios encompassing policy shifts, dynamic shifts, or a combination of both, which can not be unified in previous works.

- **Theory**

  - **Comparison between Reward and Distribution Discrepancy**: Our derivation of the surrogate policy learning objective highlights that the use of logarithmic rewards $\log r(s, a)$ is comparable to distribution discrepancies $\log(\rho_T^\pi / \rho_{\hat{T}}^{\hat{\pi}})$, as opposed to the heuristic objectives found in prior online DICE methods. For instance, AlgaeDICE (Nachum et al., 2019b) employs the objective $J(\pi) = \mathbb{E}_{(s,a) \sim d^\pi}[r(s, a) - \alpha D_f(d^\pi \| d^{\mathcal{D}})]$.

  - **Variable Substitution with Bellman Flow Constraint**: Although OMPO, AlgaeDICE and DualDICE all use variable substitution to eliminate the unknown distribution, We begin our optimisation problem by considering the Bellman flow constraint that the distribution needs to satisfy, which allows us to do variable substitutions in such a way that

$$\log r(s, a) - \alpha R(s, a, s') + \gamma \mathcal{T}^\pi Q(s, a) - Q(s, a) - \alpha y(s, a, s') = 0$$

$$\Rightarrow \quad y(s, a, s') = \frac{\log r(s, a) - \alpha R(s, a, s') + \gamma \mathcal{T}^\pi Q(s, a) - Q(s, a)}{\alpha}$$

    explaining the motivation for variable substitutions. However, previous work has simply used variable substitutions without additional specification.

  - **Offline RL with DICE Methods**: Offline RL methods like SMODICE (Ma et al., 2022) and DEMODICE (Kim et al., 2021) predominantly focus on constraining the exploration distribution of the policy to match a given distribution of offline data, aimed at avoiding out-of-distribution (OOD) issues. Consequently, even with the consideration of Bellman flow constraint, their optimization variable is $d^\pi$. In contrast, OMPO, designed for online training, uses $\pi$ as the optimization variable, with the aim of maximizing environmental rewards. The introduction of $D_f$ in our derivation naturally arises from our considerations of shifts, differentiating OMPO from these offline DICE approaches.

# E   EXPERIMENT SETTINGS

Below, we provide the environmental details for the three proposed scenarios.

**Stationary environments.**   In this scenario, we used the standard task settings from OpenAI Gym as benchmarks, including Hopper-v3, Walker2d-v3, Ant-v3, and Humanoid-v3. All methods were trained using the off-policy paradigm. Specifically, we set up a global replay buffer with a size of 1e6 for each baseline to store all historical data for policy training.

**Domain Adaption.**   For each of the four tasks, we established both source dynamics and target dynamics. In the target dynamics, we introduced significant changes, including structural modifications such as doubling the torso and foot sizes in the Hopper and Walker2d tasks, and altering mechanics such as doubling gravity and introducing a wind with a velocity of $1m/s$ in the Ant and Humanoid tasks. We divided the source dynamics into two categories, with and without the adoption of domain randomization technology.

- *Without domain randomization*: We use the standard task settings from OpenAI Gym as source dynamics, without any modification. See Table 3 for parameters comparison.

Table 3: The parameters of source dynamics without domain randomization technology

| | Source Dynamics (Without Domain Randomization) | | | | Target Dynamics | | | |
|---|---|---|---|---|---|---|---|---|
| | Torso Length | Foot Length | Gravity | Wind speed | Torso Length | Foot Length | Gravity | Wind speed |
| Hopper | 0.2 | 0.195 | - | - | 0.4 | 0.39 | - | - |
| Walker2d | 0.2 | 0.1 | - | - | 0.4 | 0.2 | - | - |
| Ant | - | - | 9.81 | 0.0 | - | - | 19.62 | 1.0 |
| Humanoid | - | - | 9.81 | 0.0 | - | - | 19.62 | 1.0 |

- *With domain randomization*: Since the principle of domain randomization is to randomise certain dynamic parameters, we apply it to the source dynamics when training the variant OMPO-DR and SAC-DR. See table 4 for the parameter settings.

Table 4: The parameters of source dynamics with domain randomization technology

| | Source Dynamics (With Domain Randomization) | | | | Target Dynamics | | | |
|---|---|---|---|---|---|---|---|---|
| | Torso Length | Foot Length | Gravity | Wind speed | Torso Length | Foot Length | Gravity | Wind speed |
| Hopper | $(0.3, 0.5)$ | $(0.29, 0.49)$ | - | - | 0.4 | 0.39 | - | - |
| Walker2d | $(0.1, 0.3)$ | $(0.05, 0.15)$ | - | - | 0.4 | 0.2 | - | - |
| Ant | - | - | $(16.62, 22.62)$ | $(0.5, 1.2)$ | - | - | 19.62 | 1.0 |
| Humanoid | - | - | $(16.62, 22.62)$ | $(0.5, 1.2)$ | - | - | 19.62 | 1.0 |

**Non-stationary environments.**   In this non-stationary dynamics scenario, dynamic variations were introduced throughout the entire training process. It's worth noting that while both non-stationary environments and domain adaptation involve dynamic shifts, the evaluation in domain adaptation is based on fixed target dynamics, whereas in non-stationary environments, the evaluation is conducted on varying dynamics. This makes the non-stationary environment evaluation more challenging. Here is a detailed description of the dynamic shifts for each task:

- Hopper task: In this task, we change the torso length $\mathcal{L}_{torso}$ and the foot length $\mathcal{L}_{foot}$ of each episode. At episode $i$, the lengths satisfy the following equations:

$$\mathcal{L}_{torso}(i) = 0.4 + 0.1 \times \sin(0.2 \times i), \quad \mathcal{L}_{foot}(i) = 0.39 + 0.1 \times \sin(0.2 \times i). \quad (30)$$

- Walker2d task: Similar to the Hopper task, the torso length $\mathcal{L}_{torso}$ and the foot length $\mathcal{L}_{foot}$ of each episode $i$ satisfy:

$$\mathcal{L}_{torso}(i) = 0.2 + 0.1 \times \sin(0.3 \times i), \quad \mathcal{L}_{foot}(i) = 0.1 + 0.05 \times \sin(0.3 \times i). \quad (31)$$

- Ant task: In the Ant task, dynamic changes occur at each time step rather than at the episode level, making it as a stochastic task. Let $i$ denote the number of episodes and $0 \le j \le 1000$

represent the time step in an episode. The values of gravity $g$ and wind speed $W$ are calculated as follows[3]:

$$g(i,j) = 14.715 + 4.905 \times \sin(0.5 \times i) + \text{rand}(-3, 3), \tag{32}$$

$$W(i,j) = 1 + 0.2 \times \sin(0.5 \times i) + \text{rand}(-0.1, 0.1). \tag{33}$$

- Humanoid task: Similar to the Ant task, gravity $g$ and wind speed $W$ in the Humanoid task are calculated as follows:

$$g(i,j) = 14.715 + 4.905 \times \sin(0.5 \times i) + \text{rand}(-3, 3), \tag{34}$$

$$W(i,j) = 1 + 0.5 \times \sin(0.5 \times i) + \text{rand}(-0.1, 0.1). \tag{35}$$

It's worth noting that when wind is introduced to the Ant and Humanoid tasks (with default density 1.2 and viscosity $2e-5$), the Ant has a larger windward area relative to the Humanoid, thus suffering the bigger drag. Therefore, this sets a smaller upper bound on the wind speed for the Ant task. This consideration helps maintain task feasibility and realism in the presence of wind dynamics.

**Stochastic manipulation tasks.** Here, we adopt two Panda Robot tasks and four robot manipulation tasks from Meta-World to evaluate OMPO. For the Panda Robot tasks, we introduce a fixed bias with a minor noise to the actions for each interaction, which is

$$\tilde{a} = a_{\text{OMPO}} + 0.05 + \text{uniform}(0, 0.01). \tag{36}$$

For the tasks from Meta-World suite, we use the original environmental settings.

---

[3]We choose 14.715 and 4.905 as the different magnifications of the original gravity $g = 9.81$, not a special design.

# F    MORE EXPERIMENTAL RESULTS

## F.1    TRAJECTORIES VISUALIZATION

We visualize the trajectories generated by OMPO on four tasks with target dynamics from domain adaption scenarios. For each trajectory, we display seven keyframes.

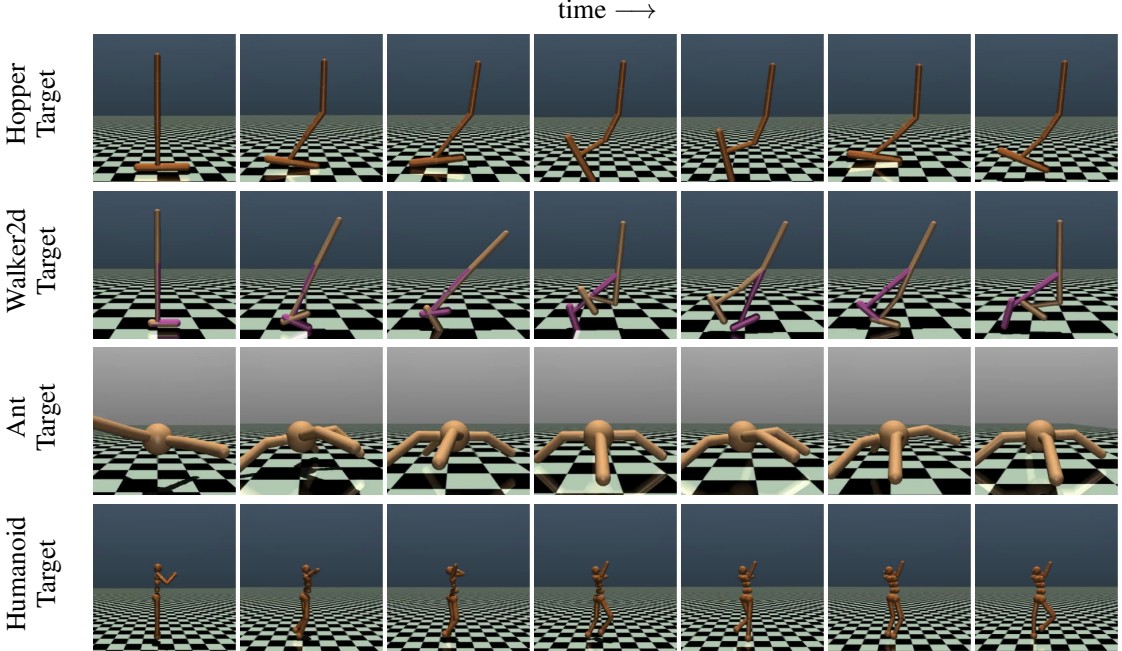

Figure 12: **Domain adaption trajectory visualizations.** Visualizations of the learned policy of OMPO on four tasks with target dynamics.

## F.2    TRANSITION OCCUPANCY DISTRIBUTION VISUALIZATION

We visualize the transition occupancy distribution $\rho_T^\pi(s, a, s')$ of the Hopper task under Non-stationary environments.

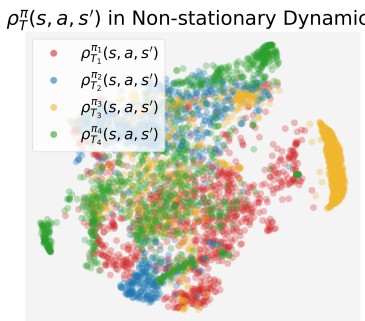

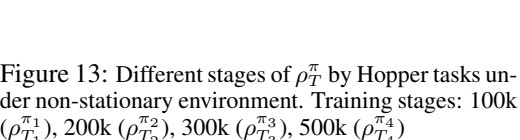

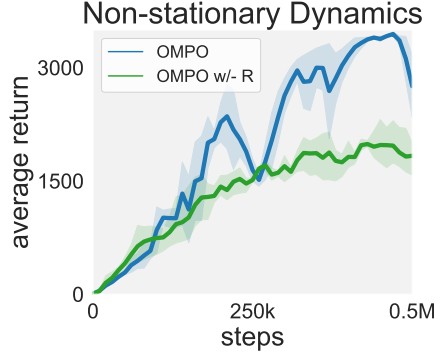

Figure 13: Different stages of $\rho_T^\pi$ by Hopper tasks under non-stationary environment. Training stages: 100k ($\rho_{T_1}^{\pi_1}$), 200k ($\rho_{T_2}^{\pi_2}$), 300k ($\rho_{T_3}^{\pi_3}$), 500k ($\rho_{T_4}^{\pi_4}$)

Figure 14: Performance comparison of OMPO and the variant of OMPO without discriminator by Hopper tasks.

## F.3 PERFORMANCE LEARNING CURVES OF STOCHASTIC ROBOT TASKS

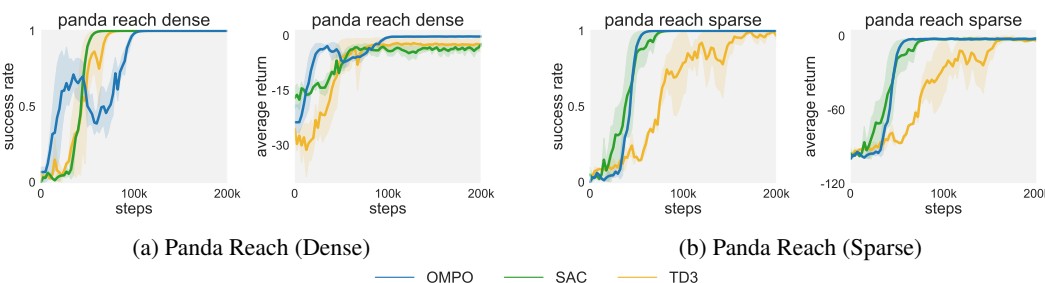

Figure 15: **Individual Meta-World tasks.** Success rate and average return of OMPO, SAC, TD3 on twelve manipulation tasks from Panda-Gym suite.

Figure 16: **Individual Panda Robot tasks.** Success rate and average return of OMPO, SAC, TD3 on two manipulation tasks from Meta-World suite.

F.4    ABLATIONS ON REWARD FUNCTION

Based on our derivation, when considering distribution discrepancies in the objective, using logarithmic rewards $\log r(s, a)$ instead of the original rewards $r(s, a)$ may be a more aligned and comparable approach. Thus, we conduct an investigation of the reward function in the surrogate objective.

- **OMPO**: the tractable problem to be solved is formulated as

$$\max_{\pi} \min_{Q(s,a)} (1 - \gamma)\mathbb{E}_{s \sim \mu_0, a \sim \pi}[Q(s, a)]$$
$$+ \alpha \mathbb{E}_{(s,a,s') \sim \rho_{\widehat{T}}^{\widehat{\pi}}} \left[ f_\star \left( \frac{\log r(s, a) - \alpha R(s, a, s') + \gamma \mathcal{T}^\pi Q(s, a) - Q(s, a)}{\alpha} \right) \right], \quad (37)$$

- **OMPO-r**: the tractable problem to be solved is formulated as

$$\max_{\pi} \min_{Q(s,a)} (1 - \gamma)\mathbb{E}_{s \sim \mu_0, a \sim \pi}[Q(s, a)]$$
$$+ \alpha \mathbb{E}_{(s,a,s') \sim \rho_{\widehat{T}}^{\widehat{\pi}}} \left[ f_\star \left( \frac{r(s, a) - \alpha R(s, a, s') + \gamma \mathcal{T}^\pi Q(s, a) - Q(s, a)}{\alpha} \right) \right], \quad (38)$$

Through the Hopper tasks under three settings, as depicted in Figure 17, we find that directly using environmental rewards in our framework, rather than in the form of logarithmic rewards, leads to performance degradation, illustrating the soundness of our theory.

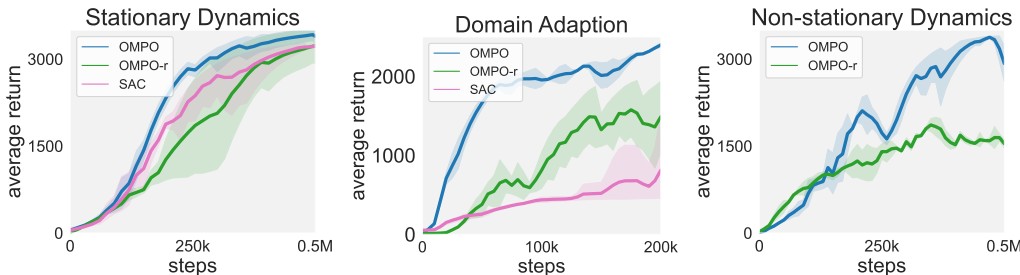

Figure 17: Performance comparison between OMPO and OMPO-r through Hopper tasks.

F.5    LONG TRAINING STEPS OF STATIONARY ENVIRONMENTS

We provide the performance comparison under 2.5M environmental steps in Figure 18. The results demonstrate that, OMPO exhibits significantly better sample efficiency and competitive convergence performance.

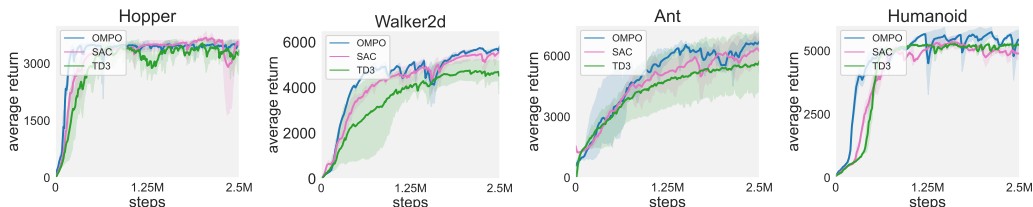

Figure 18: Performance comparison of OMPO, SAC and TD3 through 2.5M environment steps in stationary environments.

F.6    MORE SEVERE DYNAMICS SHIFTS EXPERIMENTS

To verify the robustness of OMPO in extreme cases, we conduct additional experiments in Non-stationary environments where the gravity ranges from $0.5 \sim 3$ times the original parameters.

Specifically, through Ant task and Humanoid tasks, gravity $g$ is calculated as follows:

$$g(i, j) = 17.1675 + 12.2625 \times \sin(0.5 \times i) + \text{rand}(-3, 3), \tag{39}$$

where $i$ represent the $i$-th training episode.

The results are shown in Figures 19 and 20. We find that, under much greater variations in gravity, OMPO can maintain satisfactory performance in both Ant and Humanoid tasks, while the baseline CEMRL suffers from the changes of gravity greatly, demonstrating performance degradation.

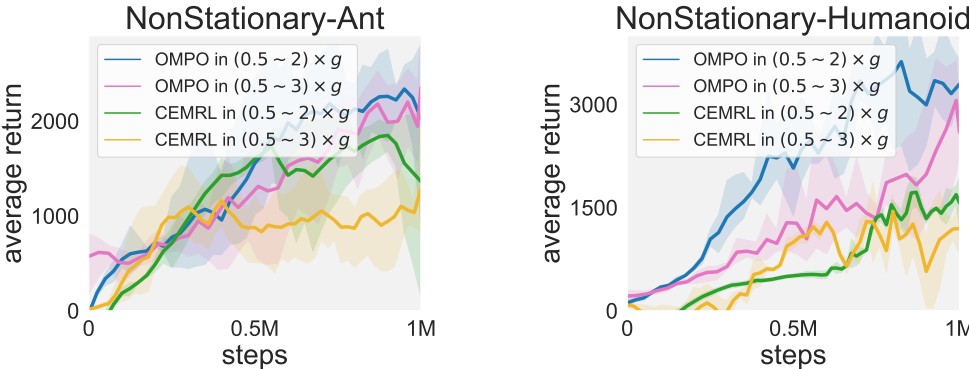

Figure 19: More Severe gravity changes in Ant tasks.  Figure 20: More Severe gravity changes in Ant tasks.

## F.7 ABLATION ON ORDER $q$ OF FENCHEL CONJUGATE

Regarding the order $q$ of Fenchel Conjugate function, we test four sets of parameters by Hopper and Walker2d tasks under non-stationary dynamics. As dispected in Figure 21, $q \in [1.2, 2]$ can yield satisfactory performance, with $q = 1.5$ showing superior results in our experiments.

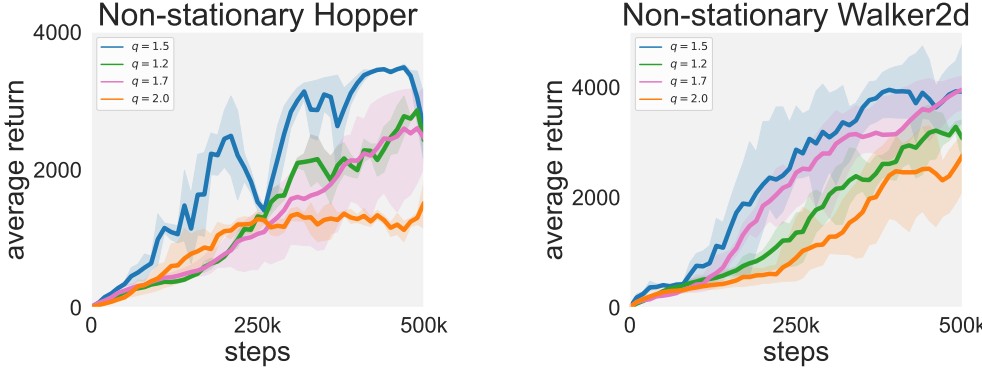

Figure 21: Ablation on the order $q$ by Hopper and Walker2d tasks under Non-stationary dynamics.

## G    COMPUTING INFRASTRUCTURE AND TRAINING TIME

We list the computing infrastructure and benchmark training times of OMPO in Table 5.

Table 5: Computing infrastructure and training time on stationary dynamics tasks (in hours).

|  | Hopper | Walker2d | Ant | Humanoid |
|---|---|---|---|---|
| CPU | Intel® Core™ i9-9900 | | | |
| GPU | NVIDIA GeForce RTX 2060 | | | |
| Training steps | 0.5M | 1.0M | 1.5M | 1.0M |
| Training time | 3.15 | 6.58 | 9.37 | 8.78 |

