# OpenReview forum: "A Unified Framework for Reinforcement Learning under Policy and Dynamic Shifts"
_ICLR.cc/2024/Conference — Submitted to ICLR 2024_

### Official Review · Reviewer_6QE6 · 2023-10-29

**Soundness:** 4 excellent
**Presentation:** 3 good
**Contribution:** 4 excellent
**Rating:** 8
**Confidence:** 4

**Summary:**

This work proposes a unified framework to address distributional shifts induced by policy changes and/or dynamic variations. This is done by firstly leveraging the concept of Transition Occupancy Distribution (TOD), which augments the state-action occupancy distribution with the next states (which “accounts” for the dynamics of the MDP). Then, the work develops a surrogate policy learning objective based on the TOD, and further reformulates such objective in a minimax optimization problem, ensuring that all terms are tractable. Finally, the work proposes a practical learning algorithm, OMPO, which implements this learning objective as an actor-critic paradigm. The work presents experiments in several continuous control environments, showing superior performance in all settings that accounts for policy or dynamic distributional shifts.

**Strengths:**

- The work approaches the challenge of handling distributional shifts in RL, which is very relevant and spawn different active areas of research. Therefore, it is well motivated.

- The proposed framework is very sound and elegant. It unifies and generalizes several challenges from Reinforcement Learning, often approached by different RL subareas (off-policy/offline RL, meta-RL, multi-task RL, sim-to-real transfer), into a single and generic learning objective.
    - It also empirically shows that OMPO can provide superior performance to several algorithms that are specialized in a single type of distributional shift, which provides strong evidence of the effectiveness from the theoretical framework and practical algorithm.

- The experimental setup is very complete. It brings all combinations of shifts (stationary environment, domain adaptation and non-stationary environments), which progressively cover all scenarios of shifts. For each case, it compares against solid and recent baselines. It also provides visualizations of the transition occupancy distributions (which explicit the problems addressed in the paper), as well as ablations for the main hyperparameters in OMPO.

- The Appendices are also rich and improve the clarity of the paper. For instance, they detail the experimental setup, hyperparameters, provide pseudocode, contrast with prior DICE methods, etc. Hence, the work looks very reproducible and mature.

**Weaknesses:**

While I do not have major concerns, I believe the paper could be improved in some directions, as detailed below:

- I believe the proof of Proposition 3 could be clearer and more didactic with a better explanation on the assumptions and on why some steps are taken. Perhaps starting with an initial “rationale” behind the proof (describing the strategy to be followed) would be helpful.
    - In fact, this could be also extended to the Section 4.2, which would help clarify some of the steps taken to arrive in the tractable learning objective (see my question below).

- First, I believe the work provided sufficient empirical evidence to support the proposed framework. Nevertheless, one question remains: does OMPO scale for harder problems? For instance, does it work in meta-RL benchmarks such as Meta-World ML10, ML45? If not, why?

- Following the previous point, the paper does not well describe the limitations of the proposed technique. It only states the challenges of tuning the buffer size but does not bring more practical information. For instance, I am curious to know about the stability of the method and how sensible it is for other learning parameters (such as those on Table 2). Additionally, it would be interesting to describe the computational resources needed to run the method for the presented benchmarks.


**Minor Concerns:**

In Section 3, while describing the MDP, I believe the paper refers to the initial state distribution by two different symbols ($\mu_0$ and $\rho_0$). Based on the rest of the paper, I believe that the $\rho_0$ should be replaced with $\mu_0$.


**Summary of the Review:**

The work provides a strong theoretical contribution by providing a framework that unifies addressing policy and dynamics shifts. The empirical part also provides a good support to the presented algorithm. The raised concerns are minor, and I recommend acceptance. Nonetheless, I am also stating medium confidence, as I do not have profound familiarity with DICE-related literature.

**Questions:**

- In the beginning of Section 4.2, the work motivates the dual reformulation due to the presence of the distribution induced by the current policy in the historical dynamics (which is unknown). Then, the paper includes the Bellman flow constraint, arriving at equations 7-8, which still depends on this unknown distribution. Could you please better justify this step? From my understanding, this seems needed to arrive at the tractable objective in Equation 12, but it is unclear if there is another justification.

- Figure 3: Would it not be possible to combine Domain Randomization with DARC? In some environments, DARC is a stronger baseline than SAC, and their combination could match or outperform OMPO-DR.



======================================== **POST-REBUTTAL** =========================================


After carefully checking other reviews and authors’ responses for all reviews, I understand that the paper improved during the rebuttal in many directions and, personally, addressed almost all of my concerns. The paper improved in clarity (more didactic and transparent in the proofs, besides new discussions) and richness in the experimental methodology (new ablations, longer runs).

I still believe that it is not clear whether OMPO would scale for more complex distributional shifts and harder environments. Nevertheless,  considering the proposed scope of this paper, I do not believe this is a major concern, as the current experiments are satisfactory to validate the proposed method. I am more confident that this paper is ready for acceptance. Therefore, I am raising my confidence (3 -> 4).

---

> ### Author Response · Authors · 2023-11-17
> **Response to Reviewer 6QE6**
>
> Thank you for your recognition and valuable suggestions. We sincerely appreciate your positive comments on the value and the results of our work. We carefully answer each of your concerns as below.
>
> > **W1&Q1: A better explanation of Proposition 3; More detail of Section 4.2; Justify the include of Bellman flow constraint.**
>
> Thanks for your suggestions. We apologize for the over brief discussion of theoretical derivation in Section 4.2, due to the length limits of the paper.
>
> * To facilitate better understanding the proof of Proposition 3, we have included a sketch of the theoretical flow for the theory of Section 4.2 in Appendix A.
> * We have provided the definition and the application of Fenchel Conjugate--the initial “rationale” of Proposition 3--in Appendix A.3 in our revision.
> * Regarding the introduction of Bellman flow constraint:
>   * Yes, we incorporate this constraint to ensure the tractability of optimization, a technique inspired by previous DICE works [1, 2, 3]. The Bellman flow constraint is essentially the definition of occupancy measure $d$, which enforces its behavior to satisfy its definition and thus will not change the original optimization problem.
>   * The key benefit of introducing this constraint is that, if we write the corresponding constraint problem into its Lagrangian dual form (Eq.(21) in the Appendix), and apply Fenchel conjugate transformation (Eqs. (19) and (22)), we can use the change of variable trick similar to DualDICE [4] and AlgaeDICE [5] (Eq.(23)) to get rid of the non-obtainable $\rho_{\widehat{T}}^{\pi}$ and convert the problem to a tractable form that only involves sampling from $\rho_{\widehat{T}}^{\widehat{\pi}}$ (Eq.(24)).
>
>
>
> > **W2: First, I believe the work provided sufficient empirical evidence to support the proposed framework. Nevertheless, one question remains: does OMPO scale for harder problems? For instance, does it work in meta-RL benchmarks such as Meta-World ML10, ML45? If not, why?**
>
> Thanks for your suggestions. Meta-World ML10, and ML45 are meta-learning tasks, which involve multiple tasks with different rewards, such as button press and drawer open. Our proposed OMPO is a single-task RL algorithm rather than a multi-task algorithm, thus is not directly applicable to these tasks. Nevertheless, we think it is possible to use OMPO as a backbone algorithm to extend it to the multi-task/meta-learning setting. We will explore it in our future works.
>
>
>
> > **W3: the stability of the method and how sensible it is for other learning parameters (such as those on Table 2). Additionally, it would be interesting to describe the computational resources needed to run the method for the presented benchmarks.**
>
> * In our ablation, we have investigated the parameters, the local buffer size $\mathcal{D}_L$ and the weighted factor $\alpha$.
> * To further address the reviewer's concern, we have conducted ablations on the order $q$ of conjugate function in Figure 21, Appendix F.7 in our revision. The results show that $q\in[1.2,2]$ can yield satisfactory performance, with $q=1.5$ showing superior results in our experiments.
> * Regarding the computation performance, although looks complex, OMPO actually does not have a high computational cost, its training time is at a similar level as SAC. We have provided computing infrastructure and training time in Appendix G in the revision.
>
>
>
> > **W4: Based on the rest of the paper, I believe that the $\rho_0$ should be replaced with $\mu_0$.**
>
> Thanks for your kindly reminder. We have corrected the typo, by replacing the $\rho_0$ as the $\mu_0$.
>
>
>
> > **Q2: Figure 3: Would it not be possible to combine Domain Randomization with DARC? In some environments, DARC is a stronger baseline than SAC, and their combination could match or outperform OMPO-DR.**
>
> Thank you for the valuable suggestion. We have conducted the experiment of DARC-DR within domain adaption tasks, which is included in Figure 3 in our revision. The results show that, the combination with domain randomization can indeed improve the performance of DARC, but OMPO-DR can still outperform DARC-DR by a large margin, indicating its superior performance. Here are the brief results of convergence performance in four tasks.
>
> | Domain Adaption | Hopper | Walker2d | Ant    | Humanoid |
> | --------------- | ------ | -------- | ------ | -------- |
> | OMPO-DR         | $2985$ | $5356$   | $4032$ | $6738$   |
> | DARC-DR         | $1743$ | $1435$   | $3187$ | $4166$   |
> | OMPO            | $2289$ | $2648$   | $3792$ | $6154$   |
> | DARC            | $894$  | $1587$   | $1261$ | $958$    |
>
> ---
>
> We would like to thank you again for the constructive comments from the reviewer. We hope our responses and refined paper can fully address your remaining concerns. If the reviewer has any further questions or comments, we are happy to address them.

---

> ### Author Response · Authors · 2023-11-17
>
> **Reference**
>
> [1] Versatile offline imitation from observations and examples via regularized state-occupancy matching. ICML 2022.
>
> [2] Mind the gap: Offline policy optimization for imperfect rewards. ICLR 2023.
>
> [3] OptiDICE: Offline Policy Optimization via Stationary Distribution Correction Estimation. ICML 2021.
>
> [4] Dualdice: Behavior-agnostic estimation of discounted stationary distribution corrections. NeurIPS 2019.
>
> [5] Algaedice: Policy gradient from arbitrary experience. arXiv 2019.

---

> ### Author Response · Authors · 2023-11-21
> **Please let us know if we have resolved your concerns**
>
> Dear reviewer,
>
> We were wondering if our responses have resolved your concerns. We hope that these clarifications and details can serve to enhance your confidence in the ratings of our work. If you have any further questions or require additional insights, we are glad to provide any necessary information to support your assessment.
>
> Best wishes!
>
> The Authors

---

> > ### Comment · Reviewer_6QE6 · 2023-11-22
> > **Thanks for addressing my concerns**
> >
> > Thank you for your responses and actions. I believe most of my concerns were addressed and I am going to raise my confidence level. Please refer to the updated version of the review.

---

> > > ### Author Response · Authors · 2023-11-22
> > > **Thank you for your inspiring reply!**
> > >
> > > Dear reviewer,
> > >
> > > We are glad that our responses helped and would like to thank you for raising the confidence of our paper. We will explore in our future research to extend OMPO to meta-learning tasks and real-world deployment.
> > >
> > > Best wishes!
> > >
> > > The authors

---

### Official Review · Reviewer_9mJF · 2023-10-31

**Soundness:** 4 excellent
**Presentation:** 4 excellent
**Contribution:** 3 good
**Rating:** 6
**Confidence:** 3

**Summary:**

This paper introduces a unified framework to tackle diverse settings of policy and dynamic shifts by performing transition occupancy matching, leading to a surrogate policy learning objective that can be cast into a tractable min-max optimization problem through employing dual formulation. Moreover, this paper conducts extensive experiments to demonstrate the efficacy of the proposed method under different policy and dynamic shifts.

**Strengths:**

1. The proposed unified framework performs consistently well in diverse settings with different policy and dynamic shifts. Notably, the authors claim they use the same set of hyperparameters for all experiments in the paper, making the results rather impressive.
2. The paper tackles the issue of policy and dynamic shifts, an important problem in deploying RL policy in real-world applications.
3. The paper is clearly written and easy to follow.

**Weaknesses:**

1. The experimental evaluations under the Stationary environments setting can be improved. For example, 1M environment steps are not usually enough when evaluating on the `Humanoid` task. I would suggest the authors provide the results of their OMPO when training for more than 2.5M environment steps and at least compare with SAC on `Walker2d`, `Ant`, and `Humanoid`.

2. The authors should provide more intuition to explain why their OMPO outperforms the baseline methods under the Stationary environments setting. Is it a consequence of incorporating $R(s, a, s')$ into the training loss? Do the authors also employ the double-Q technique for OMPO or only use a single Q?

3. The pseudo-codes provided in Algorithm 1 should provide more training details. For example, calculating Eqs (23) and Eqs (25) requires sampling $s_0$ from the distribution $\mu_0$. How does the proposed method perform this operation exactly? I suggest the authors provide more training details, at least in the appendix.

**Questions:**

1. The proposed OMPO enjoys a low variance across different random seeds in terms of performance given stationary environments, as shown in Figure 2. Can the author provide some insights into this phenomenon?

2. Eqs (23) and Eqs (25) minimize the $Q(s, a)$ specifically for $s\sim\mu_0$. Since $Q$ is parameterized by a neural network, the  $Q(s, a), s\sim\mu_0$ can be minimized spuriously low. How do the authors combat this potential training stability?

---

> ### Author Response · Authors · 2023-11-17
> **Response to Reviewer 9mJF**
>
> Thanks for your comments and suggestions. We have provided the experimental results with longer training steps, and added details in our revision. We hope our response can resolve your concerns.
>
> > **W1: The experimental evaluations under the Stationary environments setting can be improved for 2.5M.**
>
> Thanks for your suggestion. We have provided the experimental results with 2.5M environment steps in Figure 18, Appendix F.5 in our revision. We compare OMPO with SAC and TD3 through Hopper, Walker2d, Ant and Humanoid tasks. The results demonstrate that, OMPO exhibits significantly better sample efficiency and competitive convergence performance.
>
>
>
> > **W2: The authors should provide more intuition to explain why their OMPO outperforms the baseline methods under the Stationary environments setting. Is it a consequence of incorporating $R(s, a, s')$ into the training loss? Do the authors also employ the double-Q technique for OMPO or only use a single Q?**
>
> > **Q1: The proposed OMPO enjoys a low variance across different random seeds in terms of performance given stationary environments, as shown in Figure 2. Can the author provide some insights into this phenomenon?**
>
> * OMPO outperforms the baselines under the Stationary environments because it can more effectively handle policy shifts. Note that in stationary environments where $\widehat{T}=T$, objective (5) reduces to:
>
>
> $\widehat{\mathcal{J}}(\pi)=\mathbb{E}\_{(s,a,s')\sim\rho^\pi_T}\left[\log r(s,a)-\alpha\log(\rho^\pi_T/\rho^\widehat{\pi}_T)\right]-\alpha D_f\left(\rho^\pi_T\Vert\rho^\widehat{\pi}_T\right)$
>
>
> $=\mathbb{E}\_{(s,a)\sim\rho^\pi}\left[\log r(s,a)-\alpha\log(\rho^\pi/\rho^\widehat{\pi})\right]-\alpha D_f\left(\rho^\pi\Vert\rho^\widehat{\pi}\right)$
>
>
> $=\mathbb{E}\_{(s,a)\sim\rho^\pi}[\log r(s,a)]-\alpha\left[D_{KL}(\rho^\pi\Vert\rho^\widehat{\pi})+D_f(\rho^\pi\Vert\rho^\widehat{\pi})\right]$
>
> which essentially regularizes the discrepancy between on-policy occupancy $\rho^{\pi}$ and the occupancy induced by off-policy samples $\rho^\widehat{\pi}$ from the replay buffer, which helps to alleviate potential instability caused by off-policy learning [1,2]. We have provided a more detailed discussion about the performance improvement in Appendix B in our revised paper.
>
> * Yes, we also used the double-Q technique for OMPO, same as TD3 and SAC.
>
>
>
> > **W3: The pseudo-codes provided in Algorithm 1 should provide more training details. For example, calculating Eqs (23) and Eqs (25) requires sampling $s_0$ from the distribution $\mu_0$. How does the proposed method perform this operation exactly? I suggest the authors provide more training details, at least in the appendix.**
>
> Thank you very much for this valuable suggestion! To deal with $s_0\sim\mu_0$ in Eqs (23) and Eqs (25), we adopt an initial-state buffer to store initial state $s_0$ similar to the implementation of previous DICE works [3,4]. When calculating Eqs (23) and Eqs (25), we can sample $s_0$ from the initial-state buffer. We have provided this detail in Appendix C in our revision.
>
> > **Q2: Eqs (23) and Eqs (25) minimize the $Q(s, a)$ specifically for $s\sim\mu_0$. Since $Q$ is parameterized by a neural network, the $Q(s, a), s\sim\mu_0$ can be minimized spuriously low. How do the authors combat this potential training stability?**
>
> In our empirical experiments, we did not observe any numerical issue with minimizing $Q$ with $s\in \mu_0$. It should be noted that in many existing DICE-based offline RL algorithms [3, 4, 5, 6, 7], the same $\min \mathbb{E}_{s\sim \mu_0} Q(s,a)$ term is also used, and no notable computation issue is reported. We think in the online setting, the possible instability issue is even less severe than in the offline case, as the online interactive environment will randomly reset after each episode ends, potentially providing an even broader distribution for $\mu_0$.
>
> ---
>
> Thank you again for your review and comments. We hope the above responses can address your concerns and we are happy to have further discussion.

---

> ### Author Response · Authors · 2023-11-17
>
> **Reference**
>
> [1] Off-policy policy gradient with state distribution correction. Arxiv. 2019
>
> [2] State Regularized Policy Optimization on Data with Dynamics Shift. NeurIPS. 2023.
>
> [3] Algaedice: Policy gradient from arbitrary experience. arXiv preprint. 2019
>
> [4] Versatile offline imitation from observations and examples via regularized state-occupancy matching. ICML. 2022
>
> [5] Demodice: Offline imitation learning with supplementary imperfect demonstrations. ICLR 2022.
>
> [6] Mind the gap: Offline policy optimization for imperfect rewards. ICLR 2023.
>
> [7] OptiDICE: Offline Policy Optimization via Stationary Distribution Correction Estimation. ICML 2021.

---

> ### Author Response · Authors · 2023-11-21
> **We sincerely look forward to your reply.**
>
> Dear reviewer,
>
> We appreciate your comments. We were wondering if our responses and revision have resolved your concerns since only two days are left for the discussion phase. We have conducted experiments with longer environmental steps, and revised our implementation details according to your suggestions. If you have any other questions, we are also pleased to respond. We sincerely look forward to your response.
>
> Best wishes!
>
> The authors.

---

> ### Author Response · Authors · 2023-11-22
> **Awaiting your valuable feedback before deadline**
>
> Dear Reviewer 9mJF,
>
> We sincerely appreciate the time and effort you have dedicated to reviewing our work, especially during this busy period. As we approach the final 24 hours of the discussion stage, we kindly seek your feedback on our responses.
>
> In our response, we have provided the experimental results with 2.5M environment steps, clarified the performance improvement of OMPO in stationary environments, detailed the implementation of OMPO. Should you have any further questions or require additional insights, please do not hesitate to contact us. We are fully committed to supplying any necessary information to support your assessment.
>
> Best regards,
>
> The Authors

---

### Official Review · Reviewer_kdm5 · 2023-11-01

**Soundness:** 3 good
**Presentation:** 3 good
**Contribution:** 2 fair
**Rating:** 6
**Confidence:** 4

**Summary:**

The paper presents a unified framework, called Occupancy-Matching Policy Optimization (OMPO), for reinforcement learning under policy and dynamics shifts. The authors identify the challenges posed by these shifts and propose a surrogate policy learning objective that captures the transition occupancy discrepancies. They then formulate the objective as a tractable min-max optimization problem through dual reformulation. The proposed method is evaluated on benchmark environments and compared with several baselines, demonstrating its superior performance in all settings.

**Strengths:**

- The paper addresses an important problem in reinforcement learning, namely handling policy and dynamics shifts, which are common in real-world scenarios.
- The proposed OMPO framework provides a unified strategy for online RL policy learning under diverse settings of policy and dynamics shifts. The derivation of the algorithm is clear and comprehensive.
- The experimental results demonstrate that OMPO outperforms specialized baselines in various settings, showcasing its effectiveness in handling policy and dynamics shifts.

**Weaknesses:**

1. The proposed implementation is complicated. It introduces modules such as estimating the density ratio $\rho_T^\pi\left(s, a, s^{\prime}\right) / \rho_{\widehat{T}}^{\widehat{\pi}}\left(s, a, s^{\prime}\right)$ and performing min-max optimization. This can make the training unstable.
2. There is a lack of discussions on some related papers. See Q2.
3. The experiments are not thorough enough. See Q3 and Q4.

**Questions:**

1. In the related work, why do algorithms that modify the reward function require policy exploration in the source domain can provide broad data coverage? Is it due to the likelihood ratio that serves as the reward modification term? But OMPO also uses the ratio term and requires that the denominator is larger than zero.
2. Papers [1,2] also deals with the issue of dynamics shift and should be included as related works. What is the advantage of OMPO compared with these two algorithms?
3. Regarding the experiments, the change in environment parameters is limited. For example, the gravity in the target dynamics is only twice larger than that in the source dynamics. Is it possible to evaluate the algorithms with a more severe shift in dynamics?
4. How are the experiment settings related to policy shifts? It seems that all changes are made in environment parameters and related to dynamic shifts.

[1] Xue Z, Cai Q, Liu S, et al. State Regularized Policy Optimization on Data with Dynamics Shift. arXiv preprint arXiv:2306.03552, 2023.

[2] Cang C, Rajeswaran A, Abbeel P, et al. Behavioral priors and dynamics models: Improving performance and domain transfer in offline rl. arXiv preprint arXiv:2106.09119, 2021.

---

> ### Author Response · Authors · 2023-11-17
> **Response to Reviewer kdm5 (1/2)**
>
> We thank the reviewer for the constructive comments. We have conducted experiments and made modifications in our paper to address your concerns. We hope this improves our paper.
>
> > **W1: The proposed implementation is complicated. It introduces modules such as estimating the density ratio $\rho_T^\pi\left(s, a, s^{\prime}\right) / \rho_{\widehat{T}}^{\widehat{\pi}}\left(s, a, s^{\prime}\right)$ and performing min-max optimization. This can make the training unstable.**
>
> - First, the instantiation of OMPO actually is not very complicated, the only additions in OMPO as compared to typical actor-critic frameworks are a local replay buffer and a distribution discriminator.
> - Second, learning an extra discriminator for policy learning is actually quite common in methods like GAIL [1] as well as lots of DICE-based RL/IL methods [2,3,4,5]. To estimate the density ratio $\rho_T^\pi\left(s, a, s^{\prime}\right) / \rho_{\widehat{T}}^{\widehat{\pi}}\left(s, a, s^{\prime}\right)$ stably, we extend the implementation of GAIL [1] from $\rho(s,a)$ to $\rho(s,a,s')$, whose training stability has been well verified.
> - Empirically, we find the benefit of correcting policy/dynamics-shift using the extra discriminator significantly outweighs the possible instability in the discriminator. As shown in our main results in the paper, OMPO is stable and obtains superior performance, and in most cases, achieves even lower variance during policy learning.
>
>
>
> > **Q1: In the related work, why do algorithms that modify the reward function require policy exploration in the source domain can provide broad data coverage? Is it due to the likelihood ratio that serves as the reward modification term? But OMPO also uses the ratio term and requires that the denominator is larger than zero.**
>
> - We appologise for this potentially inaccurate statement in the related work. The correct sentence should be "**However, these methods often require the source domain dynamics can cover the target domain dynamics, ...**". Reward modification method like DARC [6] has an assumption that every transition with non-zero probability in the target domain will have non-zero probability in the source domain (i.e., $P_{target}(s'|s,a)>0 \Rightarrow P_{source}(s'|s,a)>0$, Eq. (1) in the DARC paper), as the computing the reward correction term $\Delta r(s,a,s')$ involves evaluating $\log(P_{target}(s'|s,a)/P_{source}(s'|s,a))$, hence require the source dynamics to be sufficiently stochastic to "cover" transitions in the target domain. This is also discussed in the Limitations section in the DARC paper.
> - Note that this problem is less severe in OMPO, as it is performing transition occupancy matching between source and target domain (i.e., between $\rho^{\pi}\_T$ and $\rho^{\hat{\pi}}\_{\hat{T}}$), which reflect long-term stationary data distribution, rather than matching at the transition dynamics level (i.e., between $P_{target}(s'|s,a)$ and $P_{source}(s'|s,a)$), thus is less restrictive.
> - We have revised this sentence in the related work in our revised paper. Thank you for pointing out this inaccurate statement.
>
>
>
> > **W2&Q2: Papers [1,2] also deals with the issue of dynamics shift and should be included as related works. What is the advantage of OMPO compared with these two algorithms?**
>
> * Thanks for the suggested papers. We have discussed them in Related Works in our revision.
> * Our proposed OMPO differs from these works, as OMPO provides a unified framework that addresses not only dynamics shifts, but also policy shifts as well as their combination. Particularly, our proposed surrogate learning objective Eq. (5) and its dual reformulation make OMPO a versatile solution that can adapt to various settings of shift scenarios.

---

> ### Author Response · Authors · 2023-11-17
> **Response to Reviewer kdm5 (2/2)**
>
> > **W3&Q3: Regarding the experiments, the change in environment parameters is limited. For example, the gravity in the target dynamics is only twice larger than that in the source dynamics. Is it possible to evaluate the algorithms with a more severe shift in dynamics?**
>
> * We thank the reviewer's valuable suggestion. In our original paper, we set the environmental parameters comparable to those in the baseline papers ($0.5\sim2$ times the original parameters). Besides, our considered dynamics shifts encompass multifaceted variations, such as simultaneous changes in wind and gravity.
> * To fully address the reviewer's concern, we have conducted additional experiments in Non-stationary environments with a broader range of $0.5\sim3$ times gravity changes. We have included the results in Figures 19 and 20, Appendix F.6 in our revision. The results show, even with a more severe shift in dynamics, OMPO can still achieve much better performance than the baselines. Here are the brief results on the average return of the Ant and Humanoid tasks in non-stationary dynamics.
>
> |                                  | Ant    | Humanoid |
> | -------------------------------- | ------ | -------- |
> | OMPO ($0.5\sim2$ times gravity)  | $2133$ | $3185$   |
> | CEMRL ($0.5\sim2$ times gravity) | $1278$ | $1546$   |
> | OMPO ($0.5\sim3$ times gravity)  | $1950$ | $2848$   |
> | CEMRL ($0.5\sim3$ times gravity) | $1265$ | $1256$   |
>
>
>
> > **W3&Q4: How are the experiment settings related to policy shifts? It seems that all changes are made in environment parameters and related to dynamic shifts.**
>
> Actually, all our methods more or less involve policy shifts, since our method as well as most baselines are off-policy algorithms, the training data sampled from the replay buffer always have some gaps with the current on-policy distribution. In Figure 2, we specifically compared our method on stationary environments (no dynamics shift) against other mainstream off-policy RL algorithms like SAC, TD3 as well as the DICE-based method AlgeaDICE, and OMPO achieves consistently better performance due to better handling of the policy shifts. In Figure 7, we also visualized the distribution discrepancy caused by policy shifts in different training stages, which shows that even in the stationary environment, the impacts of policy shifts can be clearly observed.
>
> ---
>
> We hope the explanations and additional experiments could resolve the reviewer's concerns. If the reviewer have further questions, we are happy to address them.
>
> **Reference**
>
> [1] Generative adversarial imitation learning. NeurIPS. 2016.
>
> [2] Versatile offline imitation from observations and examples via regularized state-occupancy matching. ICML 2022.
>
> [3] Offline Goal-Conditioned Reinforcement Learning via f-Advantage Regression. NeurIPS 2022.
>
> [4] Demodice: Offline imitation learning with supplementary imperfect demonstrations. ICLR 2022.
>
> [5] Mind the gap: Offline policy optimization for imperfect rewards. ICLR 2023.
>
> [6] Off-Dynamics Reinforcement Learning: Training for Transfer with Domain Classifiers. ICLR 2021.

---

> ### Author Response · Authors · 2023-11-21
> **We sincerely look forward to your reply.**
>
> Dear reviewer,
>
> We appreciate your comments. We were wondering if our responses and revision have resolved your concerns since only two days are left for the discussion phase. We hope our last reply and experiments have resolved all your concerns. If you have any other questions, we are also pleased to respond. We sincerely look forward to your response.
>
> Best wishes!
>
> The authors.

---

> ### Author Response · Authors · 2023-11-22
> **Awaiting your valuable feedback before deadline**
>
> Dear Reviewer kdm5,
>
> We sincerely appreciate the time and effort you have dedicated to reviewing our work, especially during this busy period. As we approach the final 24 hours of the discussion stage, we kindly seek your feedback on our responses.
>
> In our response, we have provided clarification on the implementation of OMPO and the existence of policy shifts in our experiments, discussed the recommended references, and conducted additional experiments on more severe dynamics shifts. Should you have any further questions or require additional insights, please do not hesitate to contact us. We are fully committed to supplying any necessary information to support your assessment.
>
> Best regards,
>
> The Authors

---

### Official Review · Reviewer_UYp5 · 2023-11-09

**Soundness:** 1 poor
**Presentation:** 2 fair
**Contribution:** 2 fair
**Rating:** 3
**Confidence:** 4

**Summary:**

In this paper, a policy learning algorithm is proposed, that is intended to learn from off policy observations, which also were collected in environments that are different from the target environment, and in particular in non-stationary environments.

It is argued that differences between environments can be expressed via  the stationary (s,a,s') distribution, which extends the standard state-action occupancy distribution (s,a) by adding the next-sate s'.  A "surrrogate" objective is proposed, that involves averaging over the (s,a,s') distribution given in the data,
which is intended to be analogous to the various DICE type off-policy methods (AlgaeDICE and others) where averaging is over the empirical occupancy (s,a).

Experiments are performed comparing the derived algorithm  to a number of current model free methods.

**Strengths:**

The problems of off-policy learning, domain adaptation and non-stationary environments are impotrtant.
The experiments presented in the paper suggest  performance improvements over the copmeting algorithms.

**Weaknesses:**

This paper lacks any theoretical justification or proper motivation of the proposed objective.

It is completely not clear why optimising the proposed objective (5) should yield good performance
on a new unseen environment, or even on the same env. off policy.  In fact, there are several logical errors in the arguments.

In more detail:

* The authors propose to construct the occupancy measure $\rho_{T}^{\pi}(s,a,s')$ and assume that observations data has such distribution. However, stantionary measures are generally ill-defined for
non-stationary envirnoments and it is not clear what this means.  Thus when the authors write $\rho_{\hat{T}}^{\pi}(s,a,s')$, it appears that they must assume the data generating env. $\hat{T}$ is a regular env, contradicting the premise of the paper.

* Even if $\rho_{\hat{T}}^{\pi}(s,a,s')$ is just computed from a finite data, it is not clear why information in it should be relevant to performance in a new environment. This might be true under some strong assumptions that are implicit, but such assumptions must be discussed. It trivially not true in the general case.


* Further, even if we only have one environment, it is not clear why the objective (5) should be related to
the standard objective $\mathcal{J}(\pi)$.  The inequlities in (2) are not tight, except in very degenerate cases. I.e. that gap between the right handside and left handside can be huge even for the optimal policy $\pi^*$.


* It is generally not clear how specifically the introduction of $s'$ helps performance on the target environment.




The encouraging experimental results do seem to indicate that there is something intersting about the proposed algorithm. However, a finished paper must provide an understanding of why the improvement happens,  or at least provide minimal theoretical grounding of the methods, both of which are absent from the current paper.

**Questions:**

Please see above.

---

> ### Author Response · Authors · 2023-11-17
> **Response to Reviewer UYp5 (1/3)**
>
> Thank you for your comments. While we respectfully disagree with some of your key points, we do appreciate the review and provide clarification to your questions and concerns below.
>
> > **W1: It is completely not clear why optimising the proposed objective (5) should yield good performance on a new unseen environment, or even on the same env.**
>
> Using off-policy [1,2] or off-dynamics [3,4] data to enhance policy learning performance and sample efficiency is widely studied in the literature. Although these samples are not from the target environment or current policy, they can transfer useful information to facilitate policy learning. Our work follows similar intuition but provides a unified and theoretical grounding solution. Compared to general RL objective (1), our proposed objective (5) penalizes policy & dynamics shifts by the terms of $\log\left(\rho^\pi_T/\rho^{\widehat{\pi}}_{\widehat{T}}\right)$ and $D_f\left(\rho^\pi\_{\widehat{T}}\|\rho^{\widehat{\pi}}\_{\widehat{T}}\right)$. In short, we penalize more on samples that have high discrepancies with the occupancy measure of the current policy $\pi$ and the target dynamics $T$, but allow to learn from samples if their discrepancies are low. For a detailed discussion,
>
> - **Same env case:** objective (5) reduces to
>
> $\widehat{\mathcal{J}}(\pi)=\mathbb{E}_{(s,a,s')\sim\rho^\pi_T}\left[\log r(s,a)-\alpha\log(\rho^\pi_T/\rho^\widehat{\pi}_T)\right]-\alpha D_f\left(\rho^\pi_T\Vert\rho^\widehat{\pi}_T\right)$
>
> $=\mathbb{E}_{(s,a)\sim\rho^\pi}\left[\log r(s,a)-\alpha\log(\rho^\pi/\rho^\widehat{\pi})\right]-\alpha D_f\left(\rho^\pi\Vert\rho^\widehat{\pi}\right)$
>
> $=\mathbb{E}\_{(s,a)\sim\rho^\pi}[\log r(s,a)]-\alpha\left[D_{KL}(\rho^\pi\Vert\rho^\widehat{\pi})+D_f(\rho^\pi\Vert\rho^\widehat{\pi})\right]$
>
> which essentially regularizes the discrepancy between on-policy occupancy $\rho^{\pi}$ and the occupancy $\rho^\widehat{\pi}$ induced by off-policy samples from the replay buffer, which helps to alleviate potential instability caused by off-policy learning [5,6].
>
> - **New unseen env case:** As mentioned in Section 4.2 and 4.3 of our paper, our proposed OMPO actually need a small number of samples collected from the target env. A key advantage of OMPO is that it can maximally leverage easily accessible data from the off-dynamics source domain environment to facilitate learning in the target domain, by matching the occupancy measure in the source domain to be close to the occupancy measure implied in the small amount of target domain data. As illustrated in Figure 6, without such regularization in SAC, the dynamics gap would negatively impact policy learning, leading to deteriorated performance.
>
>
>
> > **W2: The authors propose to construct the occupancy measure $\rho\_{T}^{\pi}(s,a,s')$ and assume that observations data has such distribution. However, stantionary measures are generally ill-defined for non-stationary envirnoments and it is not clear what this means. Thus when the authors write $\rho\_{\hat{T}}^{\pi}$, it appears that they must assume the data generating env. $\hat{T}$ is a regular env, contradicting the premise of the paper.**
>
> We think there are some misunderstandings here. As clearly illustrated in Figure 1, the non-stationary dynamics considered in our paper involve a series of different dynamics $\widehat{T}_1,\widehat{T}_2,...,\widehat{T}_h$ at different training stages, each representing a specific environment dynamics. This is a common setting considered by previous non-stationary RL studies [7, 8, 9]. We use $\widehat{T}$ to represent the transition dynamics that we encountered in the samples from the replay buffer of the **current training stage**. $\widehat{T}$ actually correspond to $\widehat{T}_1,\widehat{T}_2,...,\widehat{T}_h$ during the course of training. $\widehat{T}$ **does not** correspond to a stationary env, and we use the symbol of $\widehat{T}$ simply to keep the notations uncluttered and also provide a unified view of RL problem under policy/dynamics shifts.

---

> ### Author Response · Authors · 2023-11-17
> **Response to Reviewer UYp5 (2/3)**
>
> > **W3: Even if $\rho_{\hat{T}}^{\pi}(s,a,s')$ is just computed from a finite data, it is not clear why information in it should be relevant to performance in a new environment. This might be true under some strong assumptions that are implicit, but such assumptions must be discussed. It trivially not true in the general case.**
>
> * First, we would like to politely remind that, $\rho_{\widehat{T}}^{\pi}(s,a,s')$ actually does not need to be explicitly computed in OMPO. If the reviewer checks Step 3 of Section 4.2, as well as its proof in Appendix A.3, by leveraging Fenchel conjugate and the change of variable trick similar to DualDICE [10] and AlgaeDICE [11], we can avoid evaluating $\rho_{\widehat{T}}^{\pi}(s,a,s')$ but solve a tractable formulation Eq. (12) that only involve samples from $\rho_{\widehat{T}}^{\widehat{\pi}}$ (i.e., the off-policy/off-dynamics samples from the replay buffer).
> * Second, as we have discussed in the response to W1, using off-dynamics data to facilitate policy learning in the target domain is widely studied and acknowledged in literature [3, 4]. If the reviewer still thinks using off-dynamics samples is not relevant to policy learning in a new environment, we are happy to provide a comprehensive list of related references.
> * Lastly, as demonstrated in our empirical results in Section 5.1, OMPO achieves strong performance under environments with shifted dynamics and substantially outperforms algorithms that are specifically designed for such settings, even in some highly non-stationary environments.
>
>
> > **W4: Further, even if we only have one environment, it is not clear why the objective (5) should be related to the standard objective $\mathcal{J}(\pi)$. The inequlities in (2) are not tight, except in very degenerate cases. I.e. that gap between the right handside and left handside can be huge even for the optimal policy $\pi^{*}$.**
>
>
> * Even in a single environment, if one uses off-policy RL algorithms, the policy shifts in the off-policy training data from the replay buffer could also negatively impact performance [5, 6]. Note that, the original RL objective $\pi^*=\arg\max_{\pi}\mathbb{E}_{(s,a)\sim\rho^\pi}[r(s,a)]$ assumes the samples are from on-policy distribution $(s,a)\sim\rho^\pi$, not the off-policy distribution $(s,a)\sim\rho^{\widehat{\pi}}$. Under policy shifts, the training samples $(s,a)\sim\rho^{\widehat{\pi}}$ from a previous policy $\hat{\pi}$ have a mismatch to on-policy samples $(s,a)\sim\rho^\pi$, resulting in suboptimal performance. Thus, objective (5) penalizes the policy shifts to handle the mismatch.
> * Regarding the tightness of the inequality (2), note that under a single environment, i.e., $\widehat{T}=T$, the second term in Eq. (2) satisfies $D_{KL}\left(\rho^\pi_{\widehat{T}}\Vert\rho^\pi_{T}\right)=D_{KL}\left(\rho^\pi_T\Vert\rho^\pi_{T}\right)=0$ and right-hand side of Eq. (2) reduces to $\mathbb{E}_{(s,a,s')\sim\rho^\pi_T}[\log r(s,a)]$. This actually corresponds to solving an MDP with reward shaping using the log function. Since the log function is monotonically increasing, it does not largely change the nature of the original task.
>
>
>
> > **W5: It is generally not clear how specifically the introduction of $s'$ helps performance on the target environment.**
>
> The introduction of $s'$ is to explicitly capture the dynamics shifts between source dynamics $\widehat{T}$ and target dynamics $T$, since we need to have transition $(s,a,s')$ triples to fully specify the transition dynamics $T(s'|s,a)$. Note the transition occupancy distribution $\rho_T^{\pi}(s,a,s')$ will be different if $T$ is different, which is exactly how we our method can penalizes the dynamics shifts as in Eq.(2) (i.e., penalize with term $D_{KL}(\rho_{\widehat{T}}^\pi(s,a,s')\|\rho_T^\pi(s,a,s'))$).

---

> ### Author Response · Authors · 2023-11-17
> **Response to Reviewer UYp5 (3/3)**
>
> > **W6: The encouraging experimental results do seem to indicate that there is something interesting about the proposed algorithm. However, a finished paper must provide an understanding of why the improvement happens, or at least provide minimal theoretical grounding of the methods, both of which are absent from the current paper.**
>
> - As discussed above, our proposed objective (5) extends the general RL objective under policy & dynamics shifts, and the subsequent derivations are built upon the well-established DICE-related theories. Specifically, general RL objective $\arg\max\_{\pi}\mathbb{E}\_{(s,a)\sim\rho_\pi}[r(s,a)]$ has no treatment for policy and dynamics shifts, thus will be heavily impacted if encountering scenarios involving such shifts. In contrast, OMPO can handle these shifts carefully from our theoretical derivation using regularization terms $\log\left(\rho^\pi_T/\rho^{\widehat{\pi}}\_{\widehat{T}}\right)$ and $D_f(\left(\rho^\pi\_T\|\rho^{\widehat{\pi}}\_{\widehat{T}}\right))$, resulting in improved performance.
> - We have added a logical flow for the theoretical derivation of our method in Appendix A of our revision paper, as well as extra discussions to explain the source of performance improvement in Appendix B. Please let us know if you have any specific comments on our theoretical derivation.
>
> ---
>
> Thanks again for your comments. We hope the reviewer can reassess our work in light of these clarifications.
>
> **Reference**
>
> [1] Safe and efficient off-policy reinforcement learning. NeurIPS. 2016.
>
> [2] Data-efficient off-policy policy evaluation for reinforcement learning. ICML. 2016.
>
> [3] Domain randomization for transferring deep neural networks from simulation to the real world. IROS. 2017.
>
> [4] Transfer learning for reinforcement learning domains: A survey. JMLR. 2009.
>
> [5] Off-Policy Actor-Critic. ICML 2012.
>
> [6] Trust region policy optimization. ICML 2015.
>
> [7] Context-aware dynamics model for generalization in model-based reinforcement learning. ICLR 2022.
>
> [8] Non-stationary reinforcement learning without prior knowledge: An optimal black-box approach. COLT 2021.
>
> [9] Meta-reinforcement learning in nonstationary and dynamic environments. TPAMI, 2022.
>
> [10] Dualdice: Behavior-agnostic estimation of discounted stationary distribution corrections. NeurIPS 2019.
>
> [11] Algaedice: Policy gradient from arbitrary experience. arXiv 2019.

---

> ### Author Response · Authors · 2023-11-20
> **We sincerely look forward to your reply.**
>
> Dear reviewer,
>
> We appreciate your comments. We hope our last reply has resolved all your concerns. If you have any other questions, we are also pleased to respond. We sincerely look forward to your response.
>
> Best wishes!
>
> The authors.

---

> ### Author Response · Authors · 2023-11-21
> **Your feedback is critical to us**
>
> Dear reviewer,
>
> We were wondering if our responses and revision have resolved your concerns since only two days are left for discussion. We have added the discussion about performance improvement of OMPO compared to general RL objective, and the theoretical sketch of our derivation. Please let us know if these changes can resolve your concerns. We are eager to engage in further discussions and continue improving our work.
>
> Best regards,
>
> The Authors

---

> ### Author Response · Authors · 2023-11-22
> **Awaiting your valuable feedback before deadline**
>
> Dear Reviewer UYp5,
>
> We sincerely appreciate the time and effort you have dedicated to reviewing our work, especially during this busy period. As we approach the final 24 hours of the discussion stage, we kindly seek your feedback on our responses. Your insights are crucial in addressing any remaining concerns you may have regarding our submission.
>
> In our response, we have delved into the benefits of our proposed objective (5), clarified the definition of the occupancy measure, explained the inequalities (2) and the introduction of $s'$. Additionally, in our revision, we have incorporated a theoretical analysis of performance comparison with the general RL objective in practice, along with a theoretical sketch in our derivation.
>
> It is important to note that post-November 22nd, while we welcome any further reassessment or comments on our work at your convenience, our ability to respond may be limited. Hence, we are particularly eager to engage in a constructive discussion with you before the impending deadline. Your feedback is not only valuable to us but also essential for the final evaluation of our work.
>
> Best regards!
>
> The Authors

---

> > ### Comment · Reviewer_UYp5 · 2023-11-23
> >
> > Dear Authors,
> >
> >  Thank you for your responses.  Unfortunately, it is still not clear what is the meaning of the distribution on (s,a,s') in the case it is computed from multiple environments.
> > (by computed I do not mean one has to compute it explicitly, just defined in a way the authors prefer to define it [*]).
> >
> > In a such a case, it may correspond to a certain mixture of (s,a,s') distributions from the statinoary environments composing the non-stationary one (although not even this, in the general case). As such, there may be no policy that can approximate this mixture  in a single envirnment, and it is not clear why attempting to approximate such an average case would be useful for each envirnment indivdually.
> > Note this has nothing to do with on/off policy learning.
> >
> > I am not saying that it always unreasonable to learn such a missspecifed model as above. I am saying that the costs must be well defined, the misspecification must be stated explicitly  and analysed to understand when it is helpful.
> >
> >
> >
> > [*] The notion of "non-statinary environment" $\hat{T}$ was not formally defined in the paper, and thus also the  associated notion of  "\rho^{\pi}_{\hat{T}}" does not have a fully clear meaning.

---

> > > ### Author Response · Authors · 2023-11-23
> > >
> > > Using such notation is a common practice in prior works. And the effectiveness of our objective function has been verified from both theoretical and empirical lens.
> > >
> > > We respectfully hope you re-consider your score for the paper, since the deadline is **in last three minutes**.

---

> ### Comment · Reviewer_UYp5 · 2023-11-23
>
> This is not a question of notation. In any case, I will discuss the situation with other reviewers.

---

### Author Response · Authors · 2023-11-17
**General response**

We thank all the reviewers for the detailed and constructive comments. We have revised the paper to address the concerns of the reviewers. The summary of changes in the updated version of the paper is as follows:

1. (for Reviewer UYp5 and Reviewer 9mJF) We added the theoretical analysis of the proposed surrogate objective (5) compared to general RL objective (1), especially in stationary dynamics, to explain performance improvement, in Appendix B.
2. (for Reviewer kdm5) We discussed the suggested references in Related works.
3. (for Reviewer kdm5) We added the experiments on a range of $0.5\sim3$ times gravity in Figures 19 and 20, Appendix F.6.
4. (for Reviewer kdm5) We added the clarification in Experiments about policy shifts.
5. (for Reviewer 9mJF) We added the experimental results within 2.5M environment steps in Figure 18, Appendix F.5.
6. (for Reviewer 9mJF) We revised the pseudo-codes in Appendix C to clarify the calculation of Eqs (23) and (25).
7. (for Reviewer 6QE6) We added a theoretical sketch in Appendix A, and the explanation of the proof for Proposition 4.3.
8. (for Reviewer 6QE6) We added the parameter ablation in Figure 21, Appendix F.7.
9. (for Reviewer 6QE6) We added the comparison of DARC-DR in Figure 3.
10. (for Reviewer 6QE6) We added the computing infrastructure in Appendix G.
11. (for Reviewer 6QE6) We corrected the typo about $\rho_0$ and $\mu_0$.

---

### Meta-Review · Area_Chair_K5D1 · 2023-12-06

**Metareview:**

The paper proposes a new approach to learn from policy or dynamic shifts. The reviewers agree that the framework is appealing and the approach interesting.

The main strength of the paper is the idea of the regularization term and experiments that, for the most positive reviewer, are sufficient to show that the approach is sensible.

On the other hand, there are questions that remain unresolved even after the rebuttal, and I tend to agree that the authors didn't provide sufficent justification for their design choices.

A reviewer raised the question of why using triples rather than pairs in the regularizer, since the reward only depends on pairs. Another question was regarding the specifics of how to deal with non-stationary environments. While eventually what the authors do is clear, it is less clear _why_ they do so.

Reading the paper, I agree that the theoretical justification is lacking and that we need more discussion of the biases introduced by the method (using log rewards, using a regularizer) and what is gained through the introduction of these biases. A more throrough theoretical analysis would likely clear out the concerns/questions regarding e.g., the choice of triples or the use of log-rewards.

**Justification For Why Not Higher Score:**

A more in-depth discussion of design choices, backed up by theoretical arguments would have likely make us raise the score.

**Justification For Why Not Lower Score:**

N/A

---

### Decision · Program_Chairs · 2024-01-16

Reject